# Shallow Diffuse: Robust and Invisible Watermarking through Low-Dim Subspaces in Diffusion Models

**Wenda Li**[*]  **Huijie Zhang**[*]  **Qing Qu**
Department of Electrical Engineering & Computer Science
University of Michigan, Ann Arbor

## Abstract

The widespread use of AI-generated content from diffusion models has raised significant concerns regarding misinformation and copyright infringement. Watermarking is a crucial technique for identifying these AI-generated images and preventing their misuse. In this paper, we introduce *Shallow Diffuse*, a new watermarking technique that embeds robust and invisible watermarks into diffusion model outputs. Unlike existing approaches that integrate watermarking throughout the entire diffusion sampling process, *Shallow Diffuse* decouples these steps by leveraging the presence of a low-dimensional subspace in the image generation process. This method ensures that a substantial portion of the watermark lies in the null space of this subspace, effectively separating it from the image generation process. Our theoretical and empirical analyses show that this decoupling strategy greatly enhances the consistency of data generation and the detectability of the watermark. Extensive experiments further validate that *Shallow Diffuse* outperforms existing watermarking methods in terms of consistency.

## 1 Introduction

Diffusion models [1, 2] have recently become a new dominant family of generative models, powering various commercial applications such as Stable Diffusion [3, 4], DALL-E [5, 6], Imagen [7], Stable Audio [8] and Sora [9]. These models have significantly advanced the capabilities of text-to-image, text-to-audio, text-to-video, and multi-modal generative tasks. However, the widespread usage of AI-generated content from commercial diffusion models on the Internet has raised several serious concerns: (a) AI-generated misinformation presents serious risks to societal stability by spreading unauthorized or harmful narratives on a large scale [10–12]; (b) the memorization of training data by those models [13–17] challenges the originality of the generated content and raises potential copyright infringement issues; (c) iterative training on AI-generated content, known as model collapse [18–22] can degrade the quality and diversity of outputs over time, resulting in repetitive, biased, or low-quality generations that may reinforce misinformation and distortions in the wild Internet.

To deal with these challenges, watermarking is a crucial technique for identifying AI-generated content and mitigating its misuse. Typically, it can be applied in two main scenarios: (a) *the server scenario*, where given an initial random seed, the watermark is embedded into the image during the generation process; and (b) *the user scenario*, where given a generated image, the watermark is injected in a post-processing manner; (as shown in the left two blocks in Figure 2 top). Traditional watermarking methods [23–26] are mainly designed for the user scenario, embedding detectable watermarks directly into images with minimal modification. However, these methods are susceptible to attacks. For example, the watermarks can become undetectable with simple corruptions such as blurring on watermarked images. More recent methods considered the server scenario [27–32], enhancing robustness by integrating watermarking into the sampling process of diffusion models.

---

[*]The first and second authors contribute equally to this work.

39th Conference on Neural Information Processing Systems (NeurIPS 2025).

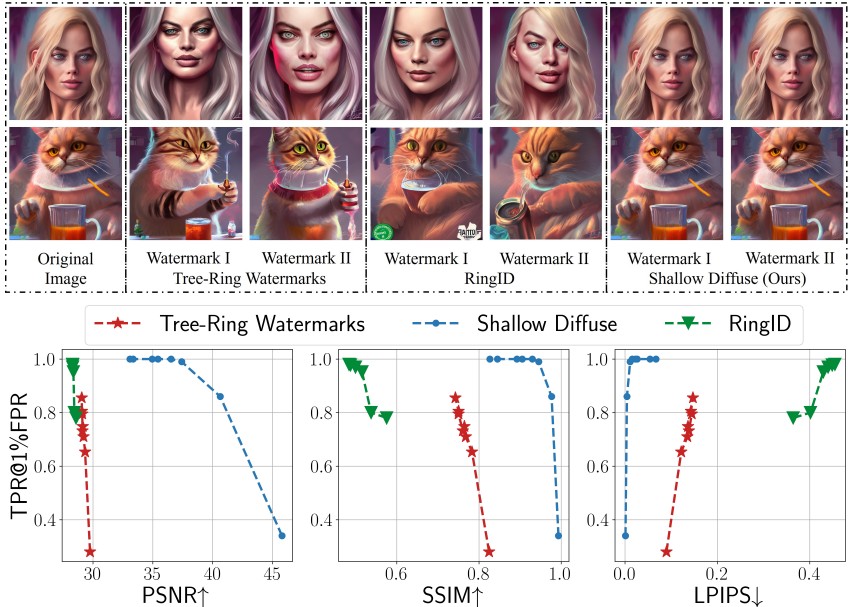

Figure 1: **Comparison between Tree-Ring Watermarks, RingID and Shallow Diffuse.** (Top) On the left are the original images, and on the right are the corresponding watermarked images generated using three techniques: Tree-Ring [29], RingID [31], and Shallow Diffuse. For each technique, we sampled watermarks using two distinct random seeds and obtained the respective watermarked images. (Bottom) Trade-off between consistency (measured by PSNR, SSIM, LPIPS) and robustness (measured by TPR@1%FPR) for Tree-Ring Watermarks, RingID, and Shallow Diffuse.

For example, recent works [29, 31] embed the watermark into the initial random seed in the Fourier domain and then sample an image from the watermarked seed. As illustrated in Figure 1, these methods frequently result in inconsistent watermarked images because they substantially distort the original Gaussian noise distribution. Moreover, since they require access to the initial random seed, it limits their use in the user scenario. To the best of our knowledge, there is no robust and consistent watermarking method suitable for both the server and user scenarios (a more detailed discussion of related works is provided in Appendix B).

To address these limitations, we proposed *Shallow Diffuse*, a robust and consistent watermarking approach that can be employed for both the server and user scenarios. In contrast to prior works [29, 31], which embed watermarks into the initial random seed and tightly couple watermarking with the sampling process, Shallow Diffuse decouples these two steps by exploiting the low-dimensional subspace structure inherent in the generation process of diffusion models [33, 34]. The key insight is that, due to the low dimensionality of the subspace, a significant portion of the watermark will lie in its null space, which effectively separates the watermarking from the sampling process (see Figure 2 for an illustration). Our theoretical and empirical analyses demonstrate that this decoupling strategy significantly improves the consistency of the watermark. Moreover, Shallow Diffuse is flexible for both server and user scenarios, with better consistency as well as independence from the initial random seed.

**Our contributions.** In summary, our proposed Shallow Diffuse offers several key advantages over existing watermarking techniques [23–32] that we highlight below:

- **Flexibility.** Watermarking via Shallow Diffuse works seamlessly under both server-side and user-side scenarios. In contrast, most of the previous methods only focus on one scenario without an easy extension to the other; see Table 1 and Table 2 for demonstrations.

- **Consistency and robustness.** By decoupling the watermarking from the sampling process, Shallow Diffuse achieves better consistency and comparable robustness. Extensive experiments (Table 1 and Table 2) support our claims, with extra ablation studies in Figure 5a and Figure 5b.

- **Provable guarantees.** The consistency and detectability of our approach are theoretically justified. Assuming a proper low-dimensional image data distribution (see Assumption 1), we rigorously establish bounds for consistency (Theorem 1) and detectability (Theorem 2).

## 2 Preliminaries

We start by reviewing the basics of diffusion models [1, 2, 35], followed by several key empirical properties that will be used in our approach: the low-rankness and local linearity of the diffusion model [33, 34].

### 2.1 Preliminaries on Diffusion Models

**Basics of diffusion models.** In general, diffusion models consist of two processes:

- *The forward diffusion process.* The forward process progressively perturbs the original data $\boldsymbol{x}_0$ to a noisy sample $\boldsymbol{x}_t$ for some integer $t \in [0, T]$ with $T \in \mathbb{Z}$. As in [1], this can be characterized by a conditional Gaussian distribution $p_t(\boldsymbol{x}_t|\boldsymbol{x}_0) = \mathcal{N}(\boldsymbol{x}_t; \sqrt{\alpha_t}\boldsymbol{x}_0, (1-\alpha_t)\mathbf{I}_d)$. Particularly, parameters $\{\alpha_t\}_{t=0}^T$ sastify: (*i*) $\alpha_0 = 1$, and thus $p_0 = p_{\text{data}}$, and (*ii*) $\alpha_T = 0$, and thus $p_T = \mathcal{N}(\mathbf{0}, \mathbf{I}_d)$.
- *The reverse sampling process.* To generate a new sample, previous works [1, 35–37] have proposed various methods to approximate the reverse process of diffusion models. Typically, these methods involve estimating the noise $\boldsymbol{\epsilon}_t$ and removing the estimated noise from $\boldsymbol{x}_t$ recursively to obtain an estimate of $\boldsymbol{x}_0$. Specifically, One sampling step of Denoising Diffusion Implicit Models (DDIM) [36] from $\boldsymbol{x}_t$ to $\boldsymbol{x}_{t-1}$ can be described as:

$$\boldsymbol{x}_{t-1} = \sqrt{\alpha_{t-1}} \underbrace{\left( \frac{\boldsymbol{x}_t - \sqrt{1-\alpha_t}\boldsymbol{\epsilon_\theta}(\boldsymbol{x}_t, t)}{\sqrt{\alpha_t}} \right)}_{:= \boldsymbol{f_{\theta,t}}(\boldsymbol{x}_t)} + \sqrt{1-\alpha_{t-1}}\boldsymbol{\epsilon_\theta}(\boldsymbol{x}_t, t), \tag{1}$$

where $\boldsymbol{\epsilon_\theta}(\boldsymbol{x}_t, t)$ is parameterized by a neural network and trained to predict the noise $\boldsymbol{\epsilon}_t$ at time $t$. From previous works [38, 39], the first term in Equation (1), defined as $\boldsymbol{f_{\theta,t}}(\boldsymbol{x}_t)$, is the *posterior mean predictor* (PMP) that predict the posterior mean $\mathbb{E}[\boldsymbol{x}_0|\boldsymbol{x}_t]$. DDIM could also be applied to a clean sample $\boldsymbol{x}_0$ and generate the corresponding noisy $\boldsymbol{x}_t$ at time $t$, named DDIM Inversion. One sampling step of DDIM inversion is similar to Equation (1), by mapping from $\boldsymbol{x}_{t-1}$ to $\boldsymbol{x}_t$. For any $t_1$ and $t_2$ with $t_2 > t_1$, we denote multi-time steps DDIM operator and its inversion as $\boldsymbol{x}_{t_1} = \texttt{DDIM}(\boldsymbol{x}_{t_2}, t_1)$ and $\boldsymbol{x}_{t_2} = \texttt{DDIM} - \texttt{Inv}(\boldsymbol{x}_{t_1}, t_2)$.

**Text-to-image (T2I) diffusion models & classifier-free guidance (CFG).** The diffusion model can be generalized from unconditional to T2I [3, 4], where the latter enables controllable image generation $\boldsymbol{x}_0$ guided by a text prompt $\boldsymbol{c}$. In more detail, when training T2I diffusion models, we optimize a conditional denoising function $\boldsymbol{\epsilon_\theta}(\boldsymbol{x}_t, t, \boldsymbol{c})$. For sampling, we employ a technique called *classifier-free guidance* (CFG) [40], which substitutes the unconditional denoiser $\boldsymbol{\epsilon_\theta}(\boldsymbol{x}_t, t)$ in Equation (1) with its conditional counterpart $\tilde{\boldsymbol{\epsilon}}_\theta(\boldsymbol{x}_t, t, \boldsymbol{c})$ that can be described as $\tilde{\boldsymbol{\epsilon}}_\theta(\boldsymbol{x}_t, t, \boldsymbol{c}) = (1 - \eta)\boldsymbol{\epsilon_\theta}(\boldsymbol{x}_t, t, \varnothing) + \eta\boldsymbol{\epsilon_\theta}(\boldsymbol{x}_t, t, \boldsymbol{c})$.. Here, $\varnothing$ denotes the empty prompt, and $\eta > 0$ denotes the strength for the classifier-free guidance. For simplification, for any $t_1$ and $t_2$ with $t_2 > t_1$, we denote multi-time steps CFG operator as $\boldsymbol{x}_{t_1} = \texttt{CFG}(\boldsymbol{x}_{t_2}, t_1, \boldsymbol{c})$. DDIM and DDIM inversion could also be generalized to T2I version, denoted by $\boldsymbol{x}_{t_1} = \texttt{DDIM}(\boldsymbol{x}_{t_2}, t_1, \boldsymbol{c})$ and $\boldsymbol{x}_{t_2} = \texttt{DDIM} - \texttt{Inv}(\boldsymbol{x}_{t_1}, t_2, \boldsymbol{c})$.

### 2.2 Local Linearity and Intrinsic Low-Dimensionality in PMP

In this work, we leverage two key properties of the PMP $\boldsymbol{f_{\theta,t}}(\boldsymbol{x}_t)$ introduced in Equation (1) for watermarking diffusion models. Parts of these properties have been previously identified in recent papers [33, 41, 42], and have been extensively analyzed in [34]. At a given timestep $t \in [0, T]$, consider the first-order Taylor expansion of the PMP $\boldsymbol{f_{\theta,t}}(\boldsymbol{x}_t + \lambda\Delta\boldsymbol{x})$ at the point $\boldsymbol{x}_t$:

$$\boxed{\boldsymbol{l_\theta}(\boldsymbol{x}_t; \lambda\Delta\boldsymbol{x}) := \boldsymbol{f_{\theta,t}}(\boldsymbol{x}_t) + \lambda\boldsymbol{J_{\theta,t}}(\boldsymbol{x}_t) \cdot \Delta\boldsymbol{x},} \tag{2}$$

where $\Delta\boldsymbol{x} \in \mathbb{S}^{d-1}$ is a perturbation direction with unit length, $\lambda \in \mathbb{R}$ is the perturbation strength, and $\boldsymbol{J_{\theta,t}}(\boldsymbol{x}_t) = \nabla_{\boldsymbol{x}_t}\boldsymbol{f_{\theta,t}}(\boldsymbol{x}_t)$ denotes the Jacobian of $\boldsymbol{f_{\theta,t}}(\boldsymbol{x}_t)$. Within a certain range of noise levels, the learned PMP $\boldsymbol{f_{\theta,t}}$ exhibits local linearity, and its Jacobian $\boldsymbol{J_{\theta,t}} \in \mathbb{R}^{d \times d}$ is low rank:

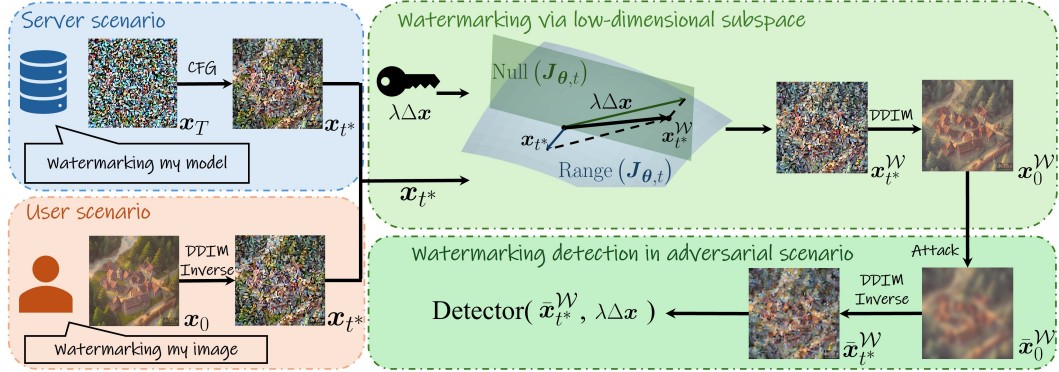

Figure 2: **Overview of Shallow Diffuse for T2I Diffusion Models.** The *server scenario* (top left) illustrates watermark embedding during generation using CFG, while the *user scenario* (bottom left) demonstrates post-generation watermark embedding via DDIM inversion. In both scenarios, the watermark is applied within a *low-dimensional subspace* (top right), where most of the watermark resides in the null space of $J_{\theta,t}$ due to its low dimensionality. The *adversarial detection* (bottom right) highlights the watermark's robustness, enabling the detector to retrieve the watermark even under adversarial attacks.

- **Low-rankness of the Jacobian $J_{\theta,t}(x_t)$.** As shown in Figure 2(a) of [34], the *rank ratio* for $t \in [0,T]$ *consistently* displays a U-shaped pattern across various network architectures and datasets: (*i*) it is close to 1 near either the pure noise $t = T$ or the clean image $t = 0$, (*ii*) $J_{\theta,t}(x_t)$ is low-rank (i.e., the numerical rank ratio is below $10^{-2}$) for all diffusion models within the range $t \in [0.2T, 0.7T]$.

- **Local linearity of the PMP $f_{\theta,t}(x_t)$.** As shown in [34, 43], the mapping $f_{\theta,t}(x_t)$ exhibits strong linearity across a large portion of the timesteps, i.e., $f_{\theta,t}(x_t + \lambda\Delta x) \approx l_\theta(x_t; \lambda\Delta x)$, a property that holds consistently true across different architectures trained on different datasets.

## 3 Watermarking by Shallow-Diffuse

This section introduces Shallow Diffuse, a training-free watermarking method designed for diffusion models. Building on the benign properties of PMP discussed in Section 2.2, we describe how to inject and detect invisible watermarks in unconditional diffusion models in Section 3.1 and Section 3.2, respectively. Algorithm 1 outlines the overall watermarking method for unconditional diffusion models. In Section 3.3, we generalize our approach to T2I diffusion models as shown in Figure 2.

### 3.1 Injecting Invisible Watermarks

Consider an unconditional diffusion model $\epsilon_\theta(x_t, t)$ as introduced in Section 2.1. Instead of injecting the watermark $\Delta x$ in the initial noise, we inject it in a particular timestep $t^* \in [0,T]$ with

$$x_{t^*}^{\mathcal{W}} = x_{t^*} + \lambda\Delta x, \qquad (3)$$

where $\lambda \in \mathbb{R}$ is the watermarking strength, $x_{t^*} = \mathtt{DDIM} - \mathtt{Inv}(x_0, t^*)$ under the user sce-

---

**Algorithm 1** Unconditional Shallow Diffuse

1: **Inject watermark:**
2: **Input**: original image $x_0$ for the user scenario (initial random seed $x_T$ for the server scenario), watermark $\lambda\Delta x$, embedding timestep $t^*$,
3: **Output**: watermarked image $x_0^{\mathcal{W}}$,
4: **if** user scenario **then**
5:      $x_{t^*} = \mathtt{DDIM} - \mathtt{Inv}(x_0, t^*)$
6: **else** server scenario
7:      $x_{t^*} = \mathtt{DDIM}(x_T, t^*)$
8: **end if**
9: $x_{t^*}^{\mathcal{W}} \leftarrow x_{t^*} + \lambda\Delta x, \; x_0^{\mathcal{W}} \leftarrow \mathtt{DDIM}(x_{t^*}^{\mathcal{W}}, 0)$
10:                ▷ Embed watermark
11: **Return:** $x_0^{\mathcal{W}}$
12:
13: **Detect watermark:**
14: **Input**: Attacked image $\bar{x}_0^{\mathcal{W}}$, watermark $\lambda\Delta x$, embedding timestep $t^*$,
15: **Output**: Distance score $\eta$,
16: $\bar{x}_{t^*}^{\mathcal{W}} \leftarrow \mathtt{DDIM} - \mathtt{Inv}(\bar{x}_0^{\mathcal{W}}, t^*)$
17: $\eta = \mathtt{Detector}(\bar{x}_{t^*}^{\mathcal{W}}, \lambda\Delta x)$
18: **Return:** $\eta$

---

nario and $x_{t^*} = \mathtt{DDIM}(x_T, t^*)$ under the server scenario. Based upon Section 2.2, we choose the timestep $t^*$ so that the Jacobian of the PMP $J_{\theta,t}(x_{t^*}) = \nabla_{x_t} f_{\theta,t}(x_{t^*})$ is *low-rank*. Moreover, based

upon the linearity of PMP discussed in Section 2.2, we approximately have

$$
\begin{aligned}
\boldsymbol{f}_{\boldsymbol{\theta},t}(\boldsymbol{x}_{t^*}^{\mathcal{W}}) \;&=\; \boldsymbol{f}_{\boldsymbol{\theta},t}(\boldsymbol{x}_{t^*}) + \lambda \underbrace{\boldsymbol{J}_{\boldsymbol{\theta},t}(\boldsymbol{x}_{t^*})\Delta\boldsymbol{x}}_{\approx \mathbf{0}} \\
&\approx\; \boldsymbol{f}_{\boldsymbol{\theta},t}(\boldsymbol{x}_{t^*}),
\end{aligned}
\tag{4}
$$

where the watermark $\Delta\boldsymbol{x}$ is designed to span the entire space $\mathbb{R}^d$ *uniformly*; a more detailed discussion on the pattern design of $\Delta\boldsymbol{x}$ is provided in Section 3.2. The key intuition for Equation (4) to hold is that, when $r_{t^*} = \text{rank}(\boldsymbol{J}_{\boldsymbol{\theta},t}(\boldsymbol{x}_{t^*}))$ is low, a significant proportion of $\lambda\Delta\boldsymbol{x}$ lies in the *null space* of $\boldsymbol{J}_{\boldsymbol{\theta},t}(\boldsymbol{x}_{t^*})$, so that $\boldsymbol{J}_{\boldsymbol{\theta},t}(\boldsymbol{x}_{t^*})\Delta\boldsymbol{x} \approx \mathbf{0}$.

Therefore, the selection of $t^*$ is based on the requirement that $\boldsymbol{f}_{\boldsymbol{\theta},t}(\boldsymbol{x}_t^*)$ is locally linear and that the rank of its Jacobian satisfies $r_t^* \ll d$. In practice, we choose $t^* = 0.3T$ based on results from the ablation study in Section 5.4. As a result, the injection in Equation (4) preserves better consistency without changing the predicted $\boldsymbol{x}_0$. In the meanwhile, it remains highly robust because any attack on $\boldsymbol{x}_0$ would remain disentangled from the watermark, so that $\lambda\Delta\boldsymbol{x}$ remains detectable.

In practice we employ the DDIM method instead of PMP for sampling high-quality images, but the above intuition still carries over to DDIM. From Equation (1), when we inject the watermark $\Delta\boldsymbol{x}$ into $\boldsymbol{x}_t^*$ as given in Equation (3), we know that

$$
\begin{aligned}
\boldsymbol{x}_{t^*-1}^{\mathcal{W}} &= \text{DDIM}(\boldsymbol{x}_{t^*}^{\mathcal{W}}, t^* - 1) \\
&\approx \sqrt{\alpha_{t^*-1}}\boldsymbol{f}_{\boldsymbol{\theta},t}(\boldsymbol{x}_{t^*}) + \frac{\sqrt{1-\alpha_{t^*-1}}}{\sqrt{1-\alpha_{t^*}}}\left(\boldsymbol{x}_{t^*} + \lambda\Delta\boldsymbol{x} - \sqrt{\alpha_{t^*}}\boldsymbol{f}_{\boldsymbol{\theta},t}(\boldsymbol{x}_{t^*})\right),
\end{aligned}
\tag{5}
$$

where the approximation follows from Equation (4). This implies that the watermark $\lambda\Delta\boldsymbol{x}$ is embedded into the DDIM sampling process entirely through the second term of Equation (5) and it decouples from the first term, which predicts $\boldsymbol{x}_0$. Therefore, similar to our analysis for PMP, the first term in Equation (5) maintains the consistency of data generation, whereas the difference in the second term, highlighted in blue, serves as a key feature for watermark detection, which we will discuss next. In Section 4, we provide rigorous proofs validating the consistency and detectability of our approach.

## 3.2 Watermark Design and Detection

Second, building on the watermark injection method described in Section 3.1, we discuss the design of the watermark pattern and the techniques for effective detection.

**Watermark pattern design.** Building on the method proposed by [29], we inject the watermark in the frequency domain to enhance robustness against adversarial attacks. Specifically, we adapt this approach by defining a watermark $\lambda\Delta\boldsymbol{x}$ for the input $\boldsymbol{x}_{t^*}$ at timestep $t^*$ as follows:

$$
\lambda\Delta\boldsymbol{x} \;:=\; \text{DFT} - \text{Inv}\left(\text{DFT}\left(\boldsymbol{x}_{t^*}\right)\odot(1-\boldsymbol{M}) + \boldsymbol{W}\odot\boldsymbol{M}\right) - \boldsymbol{x}_{t^*},
\tag{6}
$$

where the Hadamard product $\odot$ denotes the element-wise multiplication. Additionally, we have the following for Equation (6):

- **Transformation into the frequency domain.** Let $\text{DFT}(\cdot)$ and $\text{DFT} - \text{Inv}(\cdot)$ denote the forward and inverse Discrete Fourier Transform (DFT) operators, respectively. As shown in Equation (6), we first apply $\text{DFT}(\cdot)$ to transform $\boldsymbol{x}_{t^*}$ into the frequency domain, where the watermark is introduced via a mask. Finally, the modified input is transformed back into the pixel domain using $\text{DFT} - \text{Inv}(\cdot)$.

- **The mask and key of watermarks.** $\boldsymbol{M}$ is the mask used to apply the watermark in the frequency domain, as shown in the top-left of Figure 3, and $\boldsymbol{W}$ denotes the key of the watermark. Typically, the mask $\mathbf{M}$ is circular, with the white area representing 1 and the black area representing 0 in Figure 3. The mask is used to modify specific frequency bands of the image. Specifically, circular mask $M$ has a radius of 8. In the following, we discuss the design of $\boldsymbol{M}$ and $\boldsymbol{W}$ in detail.

In contrast to prior methods [29, 31], which design the mask $\boldsymbol{M}$ to modify the **low-frequency components** of the initial noise input, we construct $\boldsymbol{M}$ to target the **high-frequency components** of the image. While modifying low-frequency components is effective due to the concentration of image energy in those bands, such approaches often introduce significant visual distortion when

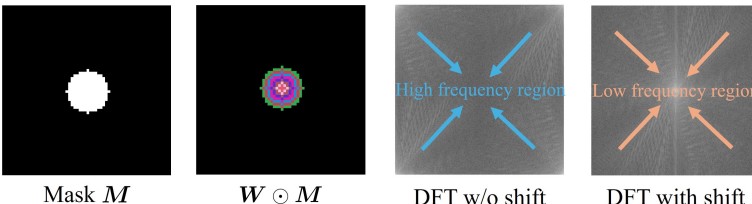

| Mask $M$ | $W \odot M$ | DFT w/o shift | DFT with shift |

Figure 3: **Visualization of Watermark Patterns**. The left two images show the circular mask $M$ and the key within the mask $M \odot W$, where the key $W$ consists of multiple rings and each sampled from the Gaussian distribution. The right two images illustrate the low- and high-frequency regions applying DFT, both before and after centering the zero frequency.

watermarks are embedded (see Figure 1 for illustration). In contrast, as shown in Figure 3, our method introduces minimal distortion by operating on the high-frequency components, which correspond to finer details and inherently contain less energy. This effect is further amplified in our case, as we apply the perturbation to $x_{t^*}$, which is closer to the clean image $x_0$, rather than to the initial noise used in [29, 31]. To isolate the high-frequency components, we apply the DFT without shifting and centering the zero-frequency component, as illustrated in the bottom-left of Figure 3.

In designing the key $W$, we follow [29]. The key $W$ is composed of multi-rings and each ring has the same value drawn from Gaussian distribution; see the top-right of Figure 3 for an illustration. Further ablation studies on the choice of $M$, $W$, and the effects of selecting low-frequency versus high-frequency regions for watermarking can be found in Table 8.

**Watermark detection.** During watermark detection, suppose we are given a watermarked image $\bar{x}_0^{\mathcal{W}}$ with certain corruptions, we apply DDIM Inversion to recover the watermarked image at timestep $t^*$, denoted as $\bar{x}_{t^*}^{\mathcal{W}} = \texttt{DDIM} - \texttt{Inv}\left(\bar{x}_0^{\mathcal{W}}, t\right)$. To detect the watermark, following [27, 29], the $\texttt{Detector}(\cdot)$ in Algorithm 1 computes the following p-value:

$$\eta = \frac{\texttt{sum}(M) \cdot ||M \odot W - M \odot \texttt{DFT}\left(\bar{x}_{t^*}^{\mathcal{W}}\right)||_F^2}{||M \odot \texttt{DFT}\left(\bar{x}_{t^*}^{\mathcal{W}}\right)||_F^2}, \tag{7}$$

where $\texttt{sum}(\cdot)$ is the summation of all elements of the matrix. Ideally, if $\bar{x}_{t^*}^{\mathcal{W}}$ is a watermarked image, $M \odot W = M \odot \texttt{DFT}\left(\bar{x}_{t^*}^{\mathcal{W}}\right)$ and $\eta = 0$. When $\bar{x}_{t^*}^{\mathcal{W}}$ is a non-watermarked image, $M \odot W \neq M \odot \texttt{DFT}\left(\bar{x}_{t^*}^{\mathcal{W}}\right)$ and $\eta > 0$. By selecting a threshold $\eta_0$, non-watermarked images satisfy $\eta > \eta_0$, while watermarked images satisfy $\eta < \eta_0$. The theoretical derivation of the p-value $\eta$ could be found in [27].

### 3.3 Extension to Text-to-Image (T2I) Diffusion Models

So far, our discussion has focused exclusively on unconditional diffusion models. Next, we show how our approach can be readily extended to T2I diffusion models, which are widely used in practice. Specifically, Figure 2 provides an overview of our method for T2I diffusion models, which can be flexibly applied to both server and user scenarios:

- **Watermark injection.** Shallow Diffuse embeds watermarks into the noise corrupted image $x_{t^*}$ at a specific timestep $t^* = 0.3T$. In the **server scenario**, given $x_T \sim \mathcal{N}(0, I_d)$ and prompt $c$, we calculate $x_{t^*} = \texttt{CFG}\left(x_T, t^*, c\right)$. In the **user scenario**, given the generated image $x_0$, we compute $x_{t^*} = \texttt{DDIM} - \texttt{Inv}\left(x_0, t^*, \varnothing\right)$, using an empty prompt $\varnothing$. Next, similar to Section 3.1, we apply DDIM to obtain the watermarked image $x_0^{\mathcal{W}} = \texttt{DDIM}\left(x_{t^*}^{\mathcal{W}}, 0, \varnothing\right)$.

- **Watermark detection.** During watermark detection, suppose we are given a watermarked image $\bar{x}_0^{\mathcal{W}}$ with certain corruptions, we apply the DDIM Inversion to recover the watermarked image at timestep $t^*$, denoted as $\bar{x}_{t^*}^{\mathcal{W}} = \texttt{DDIM} - \texttt{Inv}\left(\bar{x}_0^{\mathcal{W}}, t^*, \varnothing\right)$. We detect the watermark $\Delta x$ in $\bar{x}_{t^*}^{\mathcal{W}}$ by calculating $\eta$ in Equation (7), with detail explained in Section 3.2.

# 4 Theoretical Justification

In this section, we provide theoretical justifications for the consistency and the detectability of Shallow Diffuse for unconditional diffusion models. We begin by making the following assumptions on the watermark and the diffusion process.

**Assumption 1.** *Suppose the following holds for the PMP $\boldsymbol{f}_{\boldsymbol{\theta},t}(\boldsymbol{x}_t)$ introduced in Equation (1):*

- ***Linearity:** For any $t$ and $\Delta\boldsymbol{x} \in \mathbb{S}^{d-1}$, we always have $\boldsymbol{f}_{\boldsymbol{\theta},t}(\boldsymbol{x}_t + \lambda\Delta\boldsymbol{x}) = \boldsymbol{f}_{\boldsymbol{\theta},t}(\boldsymbol{x}_t) + \lambda\boldsymbol{J}_{\boldsymbol{\theta},t}(\boldsymbol{x}_t)\Delta\boldsymbol{x}$.*

- ***$L$-Lipschitz continuous:** we assume that $\boldsymbol{f}_{\boldsymbol{\theta},t}(\boldsymbol{x})$ is $L$-Lipschiz continuous $||\boldsymbol{J}_{\boldsymbol{\theta},t}(\boldsymbol{x})||_2 \leq L, \forall \boldsymbol{x} \in \mathbb{R}^d, t \in [0, T]$*

It should be noted that these assumptions are mild. The $L$-Lipschitz continuity is a common assumption for diffusion model analysis [44–49]. The approximated linearity have been shown in [34] with the assumption of data distribution to follow a mixture of low-rank Gaussians. For the ease of analysis, we assume exact linearity, but it can be generalized to the approximate linear case with extra perturbation analysis.

Now consider injecting a watermark $\lambda\Delta\boldsymbol{x}$ in Equation (3), where $\lambda > 0$ is a scaling factor and $\Delta\boldsymbol{x}$ is a *random* vector uniformly distributed on the unit hypersphere $\mathbb{S}^{d-1}$, i.e., $\Delta\boldsymbol{x} \sim \mathrm{U}(\mathbb{S}^{d-1})$. Then the following hold for $\boldsymbol{f}_{\boldsymbol{\theta},t}(\boldsymbol{x}_t)$.

**Theorem 1** (Consistency of the watermarks). *Suppose Assumption 1 holds and $\Delta\boldsymbol{x} \sim \mathrm{U}(\mathbb{S}^{d-1})$. Define $\hat{\boldsymbol{x}}_{0,t}^{\mathcal{W}} := \boldsymbol{f}_{\boldsymbol{\theta},t}(\boldsymbol{x}_t + \lambda\Delta\boldsymbol{x})$, $\hat{\boldsymbol{x}}_{0,t} := \boldsymbol{f}_{\boldsymbol{\theta},t}(\boldsymbol{x}_t)$. Then the $\ell_2$-norm distance between $\hat{\boldsymbol{x}}_{0,t}^{\mathcal{W}}$ and $\hat{\boldsymbol{x}}_{0,t}$ is bounded by:*

$$||\hat{\boldsymbol{x}}_{0,t}^{\mathcal{W}} - \hat{\boldsymbol{x}}_{0,t}||_2 \leq \lambda L h(r_t), \tag{8}$$

*with probability at least $1 - r_t^{-1}$. Here, $h(r_t) = \sqrt{\frac{r_t}{d} + \sqrt{\frac{18\pi^3}{d-2}\log(2r_t)}}$.*

Theorem 1 guarantees that injecting the watermark $\lambda\Delta\boldsymbol{x}$ would only change the estimation by an amount of $\lambda L h(r_t)$ with a constant probability, where $h(r_t)$ **only depends on the rank of the Jacobian** $r_t$ ($r_t \ll d$) **rather than the ambient dimension** $d$. Since $r_t$ is small, Equation (8) implies that the change in the prediction would be small. Given the relationship between PMP and DDIM in equation 1, the consistency also applies to practical use. Moreover, in the following, we show that the injected watermark remains detectable based on the second term in Equation (5).

**Theorem 2** (Detectability of the watermark). *Suppose Assumption 1 holds and $\Delta\boldsymbol{x} \sim \mathrm{U}(\mathbb{S}^{d-1})$. With $\boldsymbol{x}_t^{\mathcal{W}}$ given in Equation (3), define $\boldsymbol{x}_{t-1}^{\mathcal{W}} = \mathrm{DDIM}\left(\boldsymbol{x}_t^{\mathcal{W}}, t-1\right)$ and $\bar{\boldsymbol{x}}_t^{\mathcal{W}} = \mathrm{DDIM} - \mathrm{Inv}\left(\boldsymbol{x}_{t-1}^{\mathcal{W}}, t\right)$. The $\ell_2$-norm distance between $\tilde{\boldsymbol{x}}_t^{\mathcal{W}}$ and $\boldsymbol{x}_t^{\mathcal{W}}$ can be bounded by:*

$$||\bar{\boldsymbol{x}}_t^{\mathcal{W}} - \boldsymbol{x}_t^{\mathcal{W}}||_2 \leq \lambda L h(\max\{r_{t-1}, r_t\})[-g\left(\alpha_t, \alpha_{t-1}\right) + g\left(\alpha_{t-1}, \alpha_t\right)\left(1 - Lg\left(\alpha_t, \alpha_{t-1}\right)\right)] \tag{9}$$

*with probability at least $1 - r_t^{-1} - r_{t-1}^{-1}$. Here, $g(x, y) := \frac{\sqrt{1-y}\sqrt{x} - \sqrt{1-x}\sqrt{y}}{\sqrt{1-x}}, \forall x, y \in (0, 1)$. $h(r_t) = \sqrt{\frac{r_t}{d} + \sqrt{\frac{18\pi^3}{d-2}\log(2r_t)}}$.*

Similarly, the term $h(\max\{r_{t-1}, r_t\})$ is small because it only depends on the rank of the Jacobian $r_t$ or $r_{t-1}$ ($r_{t-1}, r_t \ll d$) rather than the ambient dimension $d$. Additionally, the term $-g\left(\alpha_t, \alpha_{t-1}\right) + g\left(\alpha_{t-1}, \alpha_t\right)\left(1 - Lg\left(\alpha_t, \alpha_{t-1}\right)\right)$ is also a small number based on the design of $\alpha_t$ for variance preserving (VP) noise scheduler [1]. Together, this implies that the difference between $\bar{\boldsymbol{x}}_t^{\mathcal{W}}$ and $\boldsymbol{x}_t^{\mathcal{W}}$ is small and $\boldsymbol{x}_t^{\mathcal{W}}$ could be recovered by $\bar{\boldsymbol{x}}_t^{\mathcal{W}}$ from one-step DDIM. Therefore, Theorem 2 implies that the injected watermark can be detected with high probability.

# 5 Experiments

In this section, we present a comprehensive set of experiments to demonstrate the robustness and consistency of *Shallow-Diffuse* across various datasets. We begin by highlighting its performance in terms of robustness and consistency in both the server scenario (Section 5.1) and the user scenario (Section 5.2). We further explore the trade-off between robustness and consistency in Section 5.3.

Table 1: **Generation quality, consistency and watermark robustness under the server scenario.** **Bold** indicates the best overall performance; Underline denotes the best among diffusion-based methods.

| Method | Generation Quality | | Generation Consistency | | | Watermark Robustness (TPR@1%FPR↑) | | | | |
|---|---|---|---|---|---|---|---|---|---|---|
| | CLIP-Score ↑ | FID ↓ | PSNR ↑ | SSIM ↑ | LPIPS ↓ | Clean | Distortion | Regeneration | Adversarial | Average |
| SD w/o WM | 0.3669 | 25.56 | - | - | - | - | - | - | - | - |
| DwtDct | 0.3641 | 25.73 | 40.32 | 0.98 | **0.01** | 0.85 | 0.35 | 0.01 | 0.42 | 0.22 |
| DwtDctSvd | 0.3629 | 26.00 | 40.19 | 0.98 | **0.01** | **1.00** | 0.74 | 0.07 | 0.01 | 0.37 |
| RivaGAN | 0.3628 | 24.60 | **40.45** | **0.99** | **0.01** | 0.99 | 0.88 | 0.05 | **0.82** | 0.54 |
| Stegastamp | 0.3410 | **24.59** | 26.70 | 0.85 | 0.08 | **1.00** | 0.99 | 0.48 | 0.05 | 0.66 |
| Stable Signature | 0.3622 | 30.86 | 32.43 | 0.95 | 0.02 | **1.00** | 0.59 | 0.19 | 0.99 | 0.48 |
| Tree-Ring | 0.3645 | 25.82 | 16.61 | 0.64 | 0.31 | **1.00** | 0.88 | 0.87 | 0.06 | 0.77 |
| RingID | 0.3637 | 27.13 | 14.27 | 0.51 | 0.42 | **1.00** | **1.00** | **1.00** | 0.33 | 0.91 |
| Gaussian Shading | 0.3663 | 26.17 | 11.04 | 0.48 | 0.54 | **1.00** | **1.00** | **1.00** | 0.47 | **0.93** |
| **Shallow Diffuse** | **0.3669** | 25.60 | 35.49 | 0.96 | 0.02 | **1.00** | **1.00** | 0.98 | 0.54 | **0.93** |

Lastly, we provide extra *multi-key identification* experiments in Appendix C.2 and ablation studies on watermark pattern design (Appendix C.3), watermarking embedded channel (Appendix C.4), watermark injecting timestep $t$ (Section 5.4) and inference steps (Appendix C.5).

**Comparison baselines.** For the server scenario, we select the following non-diffusion-based methods: DWtDct [23], DwtDctSvd [23], RivaGAN [50], StegaStamp [51]; and diffusion-based methods: Stable Signature [28], Tree-Ring Watermarks [29], RingID [31], and Gaussian Shading [30]. In the user scenario, we adopt the same baseline methods, except for Stable Signature, as this method are not suitable for this setting.

**Evaluation datasets.** We use Stable Diffusion 2-1-base [3] as the underlying model for our experiments, applying Shallow diffusion within its latent space. For the server scenario (Section 5.1), all diffusion-based methods are based on the same Stable Diffusion, with the original images $x_0$ generated from identical initial seeds $x_T$. Non-diffusion methods are applied to these same original images $x_0$ in a post-watermarking process. A total of 5000 original images are generated for evaluation in this scenario. For the user scenario (Section 5.2), we utilize the MS-COCO [52], and DiffusionDB datasets [53]. The first one is a real-world dataset, while DiffusionDB is a collection of diffusion model-generated images. From each dataset, we select 500 images for evaluation. For the remaining experiments in Section 5.3 and Appendix C, we use the server scenario and sample 100 images for evaluation.

**Evaluation metrics.** To evaluate image consistency, we use peak signal-to-noise ratio (PSNR) [54], structural similarity index measure (SSIM) [55], and Learned Perceptual Image Patch Similarity (LPIPS) [56], comparing watermarked images to their original counterparts. In the server scenario, we also assess the generation quality of the watermarked images using Contrastive Language-Image Pretraining Score (CLIP-Score) [57] and Fréchet Inception Distance (FID) [58]. To evaluate robustness, we plot the true positive rate (TPR) against the false positive rate (FPR) for the receiver operating characteristic (ROC) curve. We use the area under the curve (AUC) and TPR when FPR = 0.01 (TPR @1% FPR) as robustness metrics.

**Attacks.** Robustness is comprehensively evaluated both under clean conditions (no attacks) and with **15 types of attacks**. Following [59], we categorized them into three groups, including: distortion attack (JPEG compression, Gaussian blurring, Gaussian noise, color jitter, resize and restore, random drop, median blurring), regeneration attack (diffusion purification [60], VAE-based image compression models [61, 62], stable diffusion-based image regeneration [63], 2 times and 4 times rinsing regenerations [59]) and adversarial attack (black-box and grey-box averaging attack [64]). Here, we report only the TPR at 1% FPR for the average robustness across each group and all attacks. Detailed settings and full experiment results of these attacks are provided in Appendix C.1.

## 5.1 Server Scenario Consistency and Robustness

Table 1 compares the performance of Shallow Diffuse with other methods in the server scenario. For reference, we also apply stable diffusion to generate images from the same random seeds,

Table 2: **Generation consistency and watermark robustness under the user scenario. Bold** indicates the best overall performance; Underline denotes the best among diffusion-based methods.

| | Method | Generation Consistency | | | Watermark Robustness (AUC ↑/TPR@1%FPR↑) | | | | |
|---|---|---|---|---|---|---|---|---|---|
| | | PSNR ↑ | SSIM ↑ | LPIPS ↓ | Clean | Distortion | Regeneration | Adversarial | Average |
| COCO | SD w/o WM | 32.28 | 0.78 | 0.06 | - | - | - | - | - |
| | DwtDct | 37.88 | 0.97 | **0.02** | 0.83 | 0.54 | 0.00 | 0.82 | 0.36 |
| | DwtDctSvd | 38.06 | **0.98** | **0.02** | **1.00** | 0.76 | 0.06 | 0.00 | 0.38 |
| | RivaGAN | **40.57** | **0.98** | 0.04 | **1.00** | 0.93 | 0.05 | **1.00** | 0.59 |
| | Stegastamp | 31.88 | 0.86 | 0.08 | **1.00** | 0.97 | 0.47 | 0.26 | 0.68 |
| | Gaussian Shading | 10.17 | 0.23 | 0.65 | **1.00** | 0.99 | **1.00** | 0.47 | 0.92 |
| | Tree-Ring | 28.22 | 0.57 | 0.41 | **1.00** | 0.90 | 0.95 | 0.31 | 0.84 |
| | RingID | 12.21 | 0.38 | 0.58 | **1.00** | 0.98 | **1.00** | 0.79 | **0.96** |
| | **Shallow Diffuse** | 32.11 | 0.84 | 0.05 | **1.00** | **1.00** | 0.96 | 0.62 | 0.93 |
| DiffusionDB | SD w/o WM | 33.42 | 0.85 | 0.03 | - | - | - | - | - |
| | DwtDct | 37.77 | 0.96 | **0.02** | 0.76 | 0.34 | 0.01 | 0.78 | 0.27 |
| | DwtDctSvd | 37.84 | 0.97 | **0.02** | 1.00 | 0.74 | 0.04 | 0.00 | 0.36 |
| | RivaGAN | **40.6** | **0.98** | 0.04 | 0.98 | 0.88 | 0.04 | **0.98** | 0.56 |
| | Stegastamp | 32.03 | 0.85 | 0.08 | **1.00** | 0.96 | 0.46 | 0.26 | 0.67 |
| | Gaussian Shading | 10.61 | 0.27 | 0.63 | **1.00** | 0.99 | **1.00** | 0.46 | 0.92 |
| | Tree-Ring | 28.3 | 0.62 | 0.29 | **1.00** | 0.81 | 0.87 | 0.26 | 0.76 |
| | RingID | 12.53 | 0.45 | 0.53 | **1.00** | 0.99 | **1.00** | 0.79 | **0.97** |
| | **Shallow Diffuse** | 33.07 | 0.89 | 0.03 | **1.00** | **1.00** | 0.93 | 0.59 | 0.92 |

without adding watermarks (referred to as "SD w/o WM" in Table 1). In terms of generation quality, Shallow Diffuse achieves the best FID and CLIP scores among all diffusion-based methods. It also demonstrates superior generation consistency, achieving the highest PSNR, SSIM, and LPIPS scores. Regarding robustness, Shallow Diffuse performs comparably to Gaussian Shading and RingID, while outperforming the remaining methods. Although Gaussian Shading and RingID show similar levels of generation quality and robustness in the server scenario, their poor consistency makes them less suitable for the user scenario.

## 5.2 User Scenario Consistency and Robustness

Under the user scenario, Table 2 presents a comparison of Shallow Diffuse against other methods. In terms of consistency, Shallow Diffuse outperforms all other diffusion-based approaches. To measure the upper bound of diffusion-based methods, we apply stable diffusion with $\hat{x}_0 = \text{DDIM}(\text{DDIM} - \text{Inv}(x_0, t, \varnothing), 0, \varnothing)$, and measure the data consistency between $\hat{x}_0$ and $x_0$ (denoted in SD w/o WM in Table 2). The upper bound is constrained by errors introduced through DDIM inversion, and Shallow Diffuse comes the closest to reaching this limit. For non-diffusion-based methods, which are not affected by DDIM inversion errors, better image consistency is achievable. However, as visualized in Figure 4, Shallow Diffuse also demonstrates strong generation consistency. In terms of robustness, Shallow Diffuse performs comparably to RingID and Gaussian shading, while outperforming all other methods across both datasets. Notably, RingID and Gaussian achieve high robustness at the sacrifice of poor generation consistency (see Table 2 and Figure 4). In contrast, Shallow Diffuse is the only method that balances strong generation consistency with high watermark robustness, making it suitable for both user and server scenarios.

## 5.3 Trade-off between Consistency and Robustness

Figure 1 bottom illustrates the trade-off between consistency and robustness [2] for Shallow Diffuse and other baselines. As the radius of $M$ increases, the watermark intensity $\lambda$ also increases, reducing image consistency but improving robustness. By adjusting the radius of $M$, we plot the trade-off using PSNR, SSIM, and LPIPS against TPR@1%FPR. From Figure 1 bottom, curve of Shallow Diffuse is consistently above the curve of Tree-Ring Watermarks and RingID, demonstrating Shallow Diffuse's better consistency at the same level of robustness.

## 5.4 Ablation Study over Injecting Timesteps.

Figure 5 shows the relationship between the watermark injection timestep $t$ and both consistency and robustness [3]. Shallow Diffuse achieves optimal consistency at $t = 0.2T$ and optimal robustness

---

[2] In this experiment, we evaluate robustness against distortion attacks.

[3] In this experiment, we do not incorporate additional techniques like channel averaging or enhanced watermark patterns. Therefore, when $t = 1.0T$, the method is equivalent to Tree-Ring.

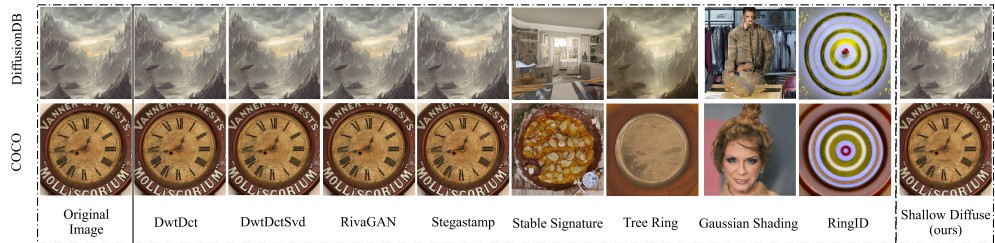

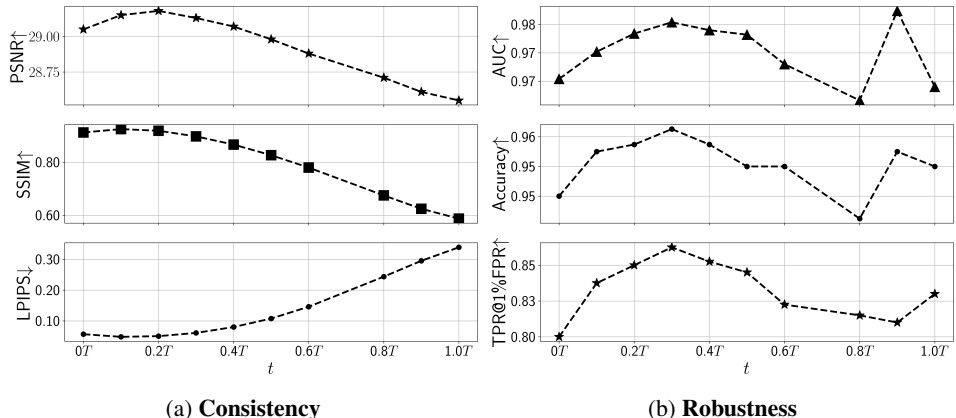

| Original Image | DwtDct | DwtDctSvd | RivaGAN | Stegastamp | Stable Signature | Tree Ring | Gaussian Shading | RingID | Shallow Diffuse (ours) |

Figure 4: **Generation consistency under user scenarios.** We compare the visualization quality of our method against DwtDct, DwtdctSvd, RivaGAN, Stegastamp, Stable Signature, Tree Ring, Gaussian Shading, and RingID across the DiffusionDB, and COCO datasets.

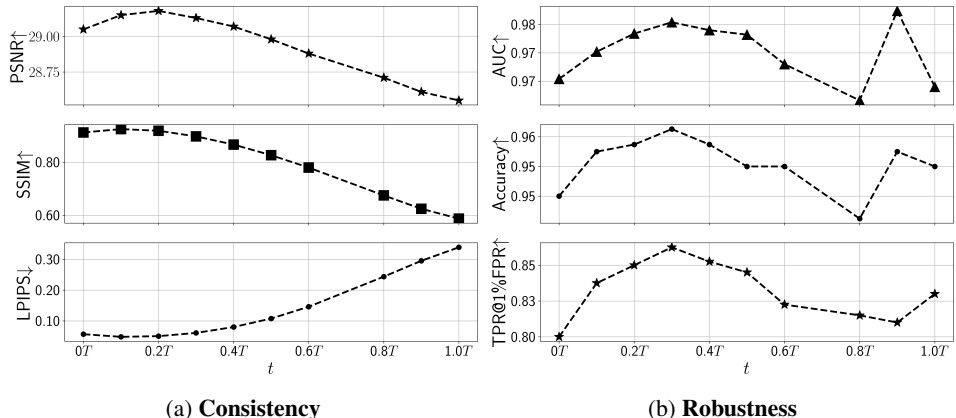

(a) **Consistency**        (b) **Robustness**

Figure 5: **Ablation study at different timestep $t$.** We evaluate the consistency and robustness under user scenarios when watermarks are injected at varying timesteps.

at $t = 0.3T$. In practice, we select $t = 0.3T$. This result aligns with the intuitive idea proposed in Section 3.1 and the theoretical analysis in Section 4: low-dimensionality enhances both data generation consistency and watermark detection robustness. However, according to [34], the optimal timestep $r_t$ for minimizing $r_t$ satisfies $t^* \in [0.5T, 0.7T]$. We believe the best consistency and robustness are not achieved at $t^*$ due to the error introduced by $\texttt{DDIM} - \texttt{Inv}$. As $t$ increases, this error grows, leading to a decline in both consistency and robustness. Therefore, the best tradeoff is reached at $t \in [0.2T, 0.3T]$, where $\boldsymbol{J}_{\boldsymbol{\theta},t}(\boldsymbol{x}_t)$ remains low-rank but $t$ is still below $t^*$. Another possible explanation is the gap between the image space and latent space in diffusion models. The rank curve in [34] is evaluated for an image-space diffusion model, whereas Shallow Diffuse operates in the latent-space diffusion model (e.g., Stable Diffusion).

## 6 Conclusion

We proposed Shallow Diffuse, a novel and flexible watermarking technique that operates seamlessly in both server-side and user-side scenarios. By decoupling the watermark from the sampling process, Shallow Diffuse achieves enhanced robustness and greater consistency. Our theoretical analysis demonstrates both the consistency and detectability of the watermarks. Extensive experiments further validate the superiority of Shallow Diffuse over existing approaches.

## 7 Acknowledgement

We acknowledge funding support from NSF CCF-2212066, NSF CCF-2212326, NSF IIS 2402950, ONR N000142512339, DARPA HR0011578254, and Google Research Scholar Award.

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

# A    Impact Statement

In this work, we introduce Shallow Diffuse, a training-free watermarking technique that hides a high-frequency signal in the low-dimensional latent subspaces of diffusion models, enabling invisible yet reliably detectable attribution for both server-side text-to-image generation and user-side post-processing. As synthetic and authentic images flooding the internet, establishing verifiable provenance is essential for copyright protection, misinformation mitigation, and scientific reproducibility. Our method preserves perceptual quality, withstands a wide range of image attacks, and requires no model retraining, making it practical for deployment. We further provide theoretical guarantees on imperceptibility and watermark recoverability, grounded in the low-rank structure of the diffusion latent space. We believe our work contributes to the development of trustworthy generative models and can inform future standards for media authentication, digital content tracking, and responsible AI deployment. While our technique could potentially be repurposed for covert signaling, we emphasize that our goal is to enhance transparency and accountability in generative AI. We encourage responsible use of this research in line with ethical guidelines and broader societal interests.

# B    Related Work

## B.1    Image Watermarking

Image watermarking has long been a crucial method for protecting intellectual property in computer vision [23–26]. Traditional techniques primarily focus on user-side watermarking, where watermarks are embedded into images post-generation. These methods [65, 66] typically operate in the frequency domain to ensure the watermarks are imperceptible. However, such watermarks remain vulnerable to adversarial attacks and can become undetectable after applying simple image manipulations like blurring.

Early deep learning-based approaches to watermarking [27, 28, 67–69] leveraged neural networks to embed watermarks. While these methods improved robustness and imperceptibility, they often suffer from high computational costs during fine-tuning and lack flexibility. Each new watermark requires additional fine-tuning or retraining, limiting their practicality.

More recently, diffusion model-based watermarking techniques have gained attraction due to their ability to seamlessly integrate watermarks during the generative process without incurring extra computational costs. Techniques such as [29–31] embed watermarks directly into the initial noise and retrieve the watermark by reversing the diffusion process. These methods enhance robustness and invisibility but are typically restricted to server-side watermarking, requiring access to the initial random seed. Moreover, the watermarks introduced by [29, 31] significantly alter the data distribution, leading to variance towards watermarks in generated outputs (as shown in Figure 1). Recent work [32] proposes embedding the watermark at an intermediate time step using adversarial optimization.

In contrast to [29, 31], our proposed shallow diffuse disentangles the watermark embedding from the generation process by leveraging the high-dimensional null space. This approach significantly improves watermark consistency while maintaining robustness. Furthermore, unlike [32], which employs adversarial optimization, our method is entirely training-free. Additionally, we provide both empirical and theoretical validation for the choice of the intermediate time step. To the best of our knowledge, this is the first training-free method that supports watermark embedding for both server-side and user-side applications while maintaining high robustness and consistency.

## B.2    Low-dimensional Subspace in Diffusion Model

In recent years, there has been growing interest in understanding deep generative models through the lens of the manifold hypothesis [70]. This hypothesis suggests that high-dimensional real-world data actually lies in latent manifolds with a low intrinsic dimension. Focusing on diffusion models, [71] empirically and theoretically shows that the approximated score function (the gradient of the log density of a noise-corrupted data distribution) in diffusion models is orthogonal to a low-dimensional subspace. Building on this, [33, 34] find that the estimated posterior mean from diffusion models lies within this low-dimensional space. Additionally, [34] discovers strong local linearity within the space, suggesting that it can be locally approximated by a linear subspace. This observation motivates our Assumption 1, where we assume the estimated posterior mean lies in a low-dimensional subspace.

Building upon these findings, [71, 72] introduce a local intrinsic dimension estimator, while [70] proposes a method for detecting out-of-domain data. [33] offers theoretical insights into how diffusion model training transitions from memorization to generalization, and [34, 41] explores the semantic basis of the subspace to achieve disentangled image editing. Unlike these previous works, our approach leverages the low-dimensional subspace for watermarking, where both empirical and theoretical evidence demonstrates that this subspace enhances robustness and consistency.

## C   Additional Experiments

### C.1   Details about Attacks

In this work, we intensively tested our method on four different watermarking attacks, both in the server scenario and in the user scenario. These watermarking attacks can be categorized into three groups, including:

- **Distortion attack**
  - JPEG compression (JPEG) with a compression rate of 25%.
  - Gaussian blurring (G.Blur) with an $8 \times 8$ filter size.
  - Gaussian noise (G.Noise) with $\sigma = 0.1$.
  - Color jitter (CJ) with brightness factor uniformly ranges between 0 and 6.
  - Resize and restore (RR). Resize to 50% of pixels and restore to original size.
  - Random drop (RD). Random drop a square with 40% of pixels.
  - Median blurring (M.Blur) with a $7 \times 7$ median filter.
- **Regeneration attack**
  - Diffusion purification [60] (DiffPure) with the purified step at 0.3T.
  - VAE-based image compression [61] (IC1) and [62] (IC2), with a quality level of 3.
  - Diffusion-based image regeneration (IR) [63].
  - Rinsing regenerations [59]) with 2 times (Rinse2x) and 4 times (Rinse4x).
- **Adversarial attack**
  - Blackbox averaging (BA) and greybox averaging (GA) watermarking removal attack [64].

Visualizations of these attacks are in Figure 6. Detailed experiments for Table 1 (Table 2) on the above attacks are reported by groups, with the distortion attack in Table 3 (Table 5) and the regeneration and adversarial attacks in Table 4 (Table 6)

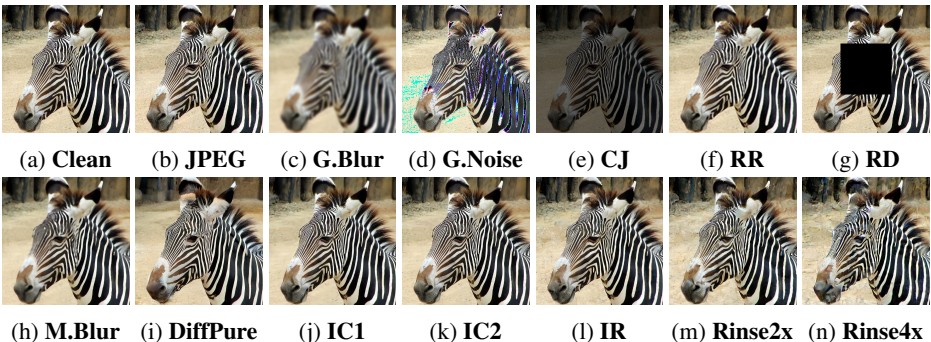

(a) **Clean** (b) **JPEG** (c) **G.Blur** (d) **G.Noise** (e) **CJ** (f) **RR** (g) **RD**

(h) **M.Blur** (i) **DiffPure** (j) **IC1** (k) **IC2** (l) **IR** (m) **Rinse2x** (n) **Rinse4x**

Figure 6: **Visualization of different attacks.**

### C.2   Multi-key Watermarking

In this section, we examine the capability of Shallow Diffuse to support multi-key watermarking. We evaluate the ability to embed multiple watermarks into the same image and detect each one independently. For this experiment, we test cases with 2, 4, 8, 16, 32 watermarks. Each watermark uses a unique ring-shaped key $W_i$ and a non-overlapped mask $M$ (part of a circle). This is a non-trivial setting as we could pre-defined the key number and non-overlapped mask $M$ for application.

Table 3: **Watermarking Robustness for distortion attacks under the server scenario.**

| Method | Watermarking Robustness (AUC ↑/TPR@1%FPR↑) | | | | | | | | |
|---|---|---|---|---|---|---|---|---|---|
| | Clean | JPEG | G.Blur | G.Noise | CJ | RR | RD | M.Blur | Distortion Average |
| DwtDct | 0.97/0.85 | 0.47/0.00 | 0.51/0.02 | 0.96/0.78 | 0.53/0.15 | 0.66/0.14 | 0.99/0.88 | 0.58/0.01 | 0.71/0.35 |
| DwtDctSvd | **1.00/1.00** | 0.64/0.10 | 0.96/0.70 | 0.99/0.99 | 0.53/0.12 | 0.99/0.99 | **1.00/1.00** | **1.00/1.00** | 0.89/0.74 |
| RivaGAN | 1.00/0.99 | 0.94/0.69 | 0.96/0.76 | 0.97/0.88 | 0.95/0.79 | 0.99/0.98 | 0.99/0.98 | 0.99/0.97 | 0.97/0.88 |
| Stegastamp | **1.00/1.00** | **1.00/1.00** | 1.00/0.95 | 0.98/0.97 | 1.00/0.97 | **1.00/1.00** | **1.00/1.00** | **1.00/1.00** | 1.00/0.99 |
| Stable Signature | **1.00/1.00** | 0.99/0.76 | 0.57/0.00 | 0.71/0.14 | 0.96/0.87 | 0.90/0.34 | **1.00/1.00** | 0.95/0.62 | 0.89/0.59 |
| Tree-Ring Watermarks | **1.00/1.00** | 0.99/0.97 | 0.98/0.98 | 0.94/0.50 | 0.96/0.67 | 0.99/0.97 | 0.99/0.94 | 0.98/0.88 |
| RingID | **1.00/1.00** | **1.00/1.00** | **1.00/1.00** | 1.00/0.99 | 0.99/0.98 | **1.00/1.00** | **1.00/1.00** | **1.00/1.00** | **1.00/1.00** |
| Gaussian Shading | **1.00/1.00** | **1.00/1.00** | **1.00/1.00** | **1.00/1.00** | **1.00/1.00** | **1.00/1.00** | **1.00/1.00** | **1.00/1.00** | **1.00/1.00** |
| **Shallow Diffuse (ours)** | **1.00/1.00** | **1.00/1.00** | **1.00/1.00** | **1.00/1.00** | **1.00/1.00** | **1.00/1.00** | **1.00/1.00** | **1.00/1.00** | **1.00/1.00** |

Table 4: **Watermarking Robustness for regeneration and adversarial attacks under the server scenario.**

| Method | Watermarking Robustness (AUC ↑/TPR@1%FPR↑) | | | | | | | | | |
|---|---|---|---|---|---|---|---|---|---|---|
| | DiffPure | IC1 | IC2 | IR | Rinse2x | Rinse4x | Regeneration Average | BA | GA | Adversarial Average |
| DwtDct | 0.50/0.00 | 0.52/0.01 | 0.49/0.00 | 0.50/0.00 | 0.77/0.04 | 0.80/0.03 | 0.60/0.01 | 0.27/0.00 | 0.99/0.84 | 0.63/0.42 |
| DwtDctSvd | 0.51/0.02 | 0.73/0.03 | 0.68/0.04 | 0.70/0.07 | 0.78/0.18 | 0.78/0.10 | 0.70/0.07 | 0.86/0.02 | 0.17/0.00 | 0.52/0.01 |
| RivaGAN | 0.73/0.16 | 0.65/0.03 | 0.63/0.04 | 0.56/0.00 | 0.64/0.03 | 0.58/0.02 | 0.63/0.05 | 0.94/0.64 | 1.00/1.00 | 0.97/0.82 |
| Stegastamp | 0.81/0.29 | 1.00/0.97 | 1.00/0.99 | 0.90/0.43 | 0.75/0.13 | 0.67/0.06 | 0.85/0.48 | 0.63/0.03 | 0.68/0.06 | 0.66/0.05 |
| Stable Signature | 0.54/0.01 | 0.93/0.58 | 0.91/0.50 | 0.67/0.02 | 0.64/0.01 | 0.54/0.01 | 0.71/0.19 | 1.00/0.98 | 1.00/1.00 | **1.00/0.99** |
| Tree-Ring Watermarks | 0.98/0.73 | 0.99/0.97 | 0.99/0.98 | 0.99/0.92 | 0.98/0.88 | 0.96/0.75 | 0.98/0.87 | 0.16/0.08 | 0.05/0.03 | 0.11/0.06 |
| RingID | **1.00/1.00** | **1.00/1.00** | **1.00/1.00** | **1.00/1.00** | **1.00/1.00** | 1.00/0.99 | **1.00/1.00** | 0.44/0.35 | 0.40/0.31 | 0.42/0.33 |
| Gaussian Shading | **1.00/1.00** | **1.00/1.00** | **1.00/1.00** | **1.00/1.00** | **1.00/1.00** | **1.00/1.00** | **1.00/1.00** | 0.53/0.48 | 0.52/0.47 | 0.53/0.47 |
| **Shallow Diffuse (ours)** | **1.00/1.00** | **1.00/1.00** | **1.00/1.00** | 1.00/0.99 | **1.00/1.00** | 0.99/0.90 | 1.00/0.98 | 0.57/0.45 | 0.70/0.63 | 0.64/0.54 |

Table 5: **Watermarking Robustness for distortion attacks under the user scenario.**

| Method | Watermarking Robustness (AUC ↑/TPR@1%FPR↑) | | | | | | | | |
|---|---|---|---|---|---|---|---|---|---|
| | Clean | JPEG | G.Blur | G.Noise | CJ | RR | RD | M.Blur | Average |
| **COCO Dataset** | | | | | | | | | |
| DwtDct | 0.98/0.83 | 0.50/0.01 | 0.50/0.00 | 0.97/0.81 | 0.54/0.14 | 0.67/0.17 | 0.99/0.93 | 0.59/0.05 | 0.64/0.54 |
| DwtDctSvd | **1.00/1.00** | 0.64/0.13 | 0.98/0.83 | 0.99/0.99 | 0.54/0.13 | **1.00/1.00** | **1.00/1.00** | **1.00/1.00** | 0.89/0.76 |
| RivaGAN | **1.00/1.00** | 0.97/0.86 | 0.98/0.86 | 0.99/0.94 | 0.96/0.82 | 1.00/1.00 | 1.00/1.00 | **1.00/1.00** | 0.99/0.93 |
| Stegastamp | **1.00/1.00** | **1.00/1.00** | 0.99/0.90 | 0.90/0.87 | 1.00/0.98 | 1.00/0.99 | 1.00/0.99 | **1.00/1.00** | 0.99/0.97 |
| Tree-Ring Watermarks | **1.00/1.00** | 0.99/0.87 | 0.99/0.86 | **1.00/1.00** | 0.88/0.49 | **1.00/1.00** | **1.00/1.00** | **1.00/1.00** | 0.98/0.90 |
| RingID | **1.00/1.00** | **1.00/1.00** | **1.00/1.00** | 0.98/0.86 | **1.00/0.99** | **1.00/1.00** | **1.00/1.00** | **1.00/1.00** | 1.00/0.98 |
| Gaussian Shading | **1.00/1.00** | **1.00/1.00** | **1.00/1.00** | **1.00/1.00** | 1.00/0.95 | **1.00/1.00** | **1.00/1.00** | **1.00/1.00** | 1.00/0.99 |
| **Shallow Diffuse (ours)** | **1.00/1.00** | **1.00/1.00** | **1.00/1.00** | **1.00/1.00** | **1.00/0.99** | **1.00/1.00** | **1.00/1.00** | **1.00/1.00** | **1.00/1.00** |
| **DiffusionDB Dataset** | | | | | | | | | |
| DwtDct | 0.96/0.76 | 0.47/0.002 | 0.51/0.018 | 0.96/0.78 | 0.53/0.15 | 0.66/0.14 | 0.99/0.88 | 0.58/0.01 | 0.71/0.34 |
| DwtDctSvd | **1.00/1.00** | 0.64/0.10 | 0.96/0.70 | 0.99/0.99 | 0.53/0.12 | **1.00/1.00** | **1.00/1.00** | **1.00/1.00** | 0.89/0.74 |
| RivaGAN | 1.00/0.98 | 0.94/0.69 | 0.96/0.76 | 0.97/0.88 | 0.95/0.79 | 1.00/0.98 | 0.99/0.98 | **1.00/1.00** | 0.98/0.88 |
| Stegastamp | **1.00/1.00** | **1.00/1.00** | 0.99/0.88 | 0.91/0.89 | 1.00/0.99 | 1.00/0.97 | **1.00/1.00** | 1.00/0.96 | 0.99/0.96 |
| Tree-Ring Watermarks | **1.00/1.00** | 0.99/0.68 | 0.94/0.62 | **1.00/1.00** | 0.84/0.15 | **1.00/1.00** | **1.00/1.00** | **1.00/1.00** | 0.97/0.81 |
| RingID | **1.00/1.00** | **1.00/1.00** | **1.00/1.00** | 0.98/0.86 | 1.00/0.98 | **1.00/1.00** | **1.00/1.00** | **1.00/1.00** | 1.00/0.98 |
| Gaussian Shading | **1.00/1.00** | **1.00/1.00** | **1.00/1.00** | **1.00/1.00** | 0.99/0.96 | **1.00/1.00** | **1.00/1.00** | **1.00/1.00** | 1.00/0.99 |
| **Shallow Diffuse (ours)** | **1.00/1.00** | 1.00/0.99 | 1.00/0.99 | **1.00/1.00** | **1.00/1.00** | **1.00/1.00** | **1.00/1.00** | **1.00/1.00** | **1.00/1.00** |

Table 6: **Watermarking Robustness for regeneration and adversarial attacks under the user scenario.**

| Method | Watermarking Robustness (AUC ↑/TPR@1%FPR↑) | | | | | | | | | |
|---|---|---|---|---|---|---|---|---|---|---|
| | DiffPure | IC1 | IC2 | IR | Rinse2x | Rinse4x | Regeneration Average | BA | GA | Adversarial Average |
| **COCO Dataset** | | | | | | | | | | |
| DwtDct | 0.46/0.00 | 0.49/0.00 | 0.49/0.01 | 0.46/0.00 | 0.61/0.00 | 0.65/0.01 | 0.53/0.00 | 0.97/0.80 | 0.96/0.84 | 0.97/0.82 |
| DwtDctSvd | 0.50/0.01 | 0.70/0.05 | 0.64/0.04 | 0.68/0.07 | 0.72/0.08 | 0.69/0.08 | 0.66/0.06 | 0.79/0.00 | 0.49/0.00 | 0.64/0.00 |
| RivaGAN | 0.63/0.02 | 0.68/0.05 | 0.66/0.04 | 0.75/0.15 | 0.68/0.03 | 0.69/0.05 | 0.69/0.05 | **1.00/1.00** | **1.00/1.00** | **1.00/1.00** |
| Stegastamp | 0.81/0.27 | 1.00/0.95 | 1.00/0.95 | 0.85/0.28 | 0.78/0.23 | 0.69/0.16 | 0.86/0.47 | 0.73/0.23 | 0.71/0.28 | 0.72/0.26 |
| Tree-Ring Watermarks | **1.00/1.00** | **1.00/1.00** | **1.00/1.00** | **1.00/1.00** | 0.99/0.92 | 0.98/0.78 | 1.00/0.95 | 0.60/0.39 | 0.46/0.23 | 0.53/0.31 |
| RingID | **1.00/1.00** | **1.00/1.00** | **1.00/1.00** | **1.00/1.00** | **1.00/1.00** | **1.00/1.00** | **1.00/1.00** | 0.75/0.59 | **1.00/1.00** | 0.88/0.79 |
| Gaussian Shading | **1.00/1.00** | **1.00/1.00** | **1.00/1.00** | **1.00/1.00** | **1.00/1.00** | **1.00/1.00** | **1.00/1.00** | 0.53/0.48 | 0.52/0.47 | 0.53/0.47 |
| **Shallow Diffuse (ours)** | 0.99/0.86 | 1.00/0.99 | 0.99/0.97 | **1.00/1.00** | **1.00/1.00** | 1.00/0.93 | 1.00/0.96 | 0.70/0.62 | 0.70/0.62 | 0.70/0.62 |
| **DiffusionDB Dataset** | | | | | | | | | | |
| DwtDct | 0.50/0.00 | 0.52/0.01 | 0.49/0.00 | 0.50/0.00 | 0.64/0.00 | 0.66/0.02 | 0.55/0.01 | 0.97/0.79 | 0.97/0.77 | 0.97/0.78 |
| DwtDctSvd | 0.51/0.02 | 0.73/0.03 | 0.68/0.04 | 0.70/0.07 | 0.73/0.07 | 0.66/0.02 | 0.67/0.04 | 0.77/0.00 | 0.39/0.00 | 0.58/0.00 |
| RivaGAN | 0.56/0.00 | 0.65/0.03 | 0.63/0.04 | 0.73/0.16 | 0.70/0.02 | 0.63/0.01 | 0.65/0.04 | 1.00/0.98 | **1.00/0.99** | **1.00/0.98** |
| Stegastamp | 0.83/0.28 | 1.00/0.91 | 1.00/0.93 | 0.85/0.40 | 0.78/0.13 | 0.68/0.11 | 0.86/0.46 | 0.69/0.21 | 0.71/0.30 | 0.70/0.26 |
| Tree-Ring Watermarks | 0.99/0.99 | 0.99/0.99 | 0.99/0.98 | 0.96/0.80 | 0.98/0.81 | 0.95/0.54 | 0.98/0.87 | 0.51/0.32 | 0.38/0.20 | 0.45/0.26 |
| RingID | **1.00/1.00** | **1.00/1.00** | **1.00/1.00** | **1.00/1.00** | **1.00/1.00** | 1.00/0.99 | **1.00/1.00** | 0.71/0.58 | **1.00/1.00** | 0.85/0.79 |
| Gaussian Shading | 1.00/0.99 | 1.00/0.99 | **1.00/1.00** | **1.00/1.00** | **1.00/1.00** | **1.00/1.00** | **1.00/1.00** | 0.50/0.46 | 0.50/0.46 | 0.50/0.46 |
| **Shallow Diffuse (ours)** | 0.96/0.90 | 0.96/0.92 | 0.97/0.93 | 0.98/0.96 | 1.00/0.98 | 0.98/0.88 | 0.97/0.93 | 0.66/0.58 | 0.68/0.60 | 0.67/0.59 |

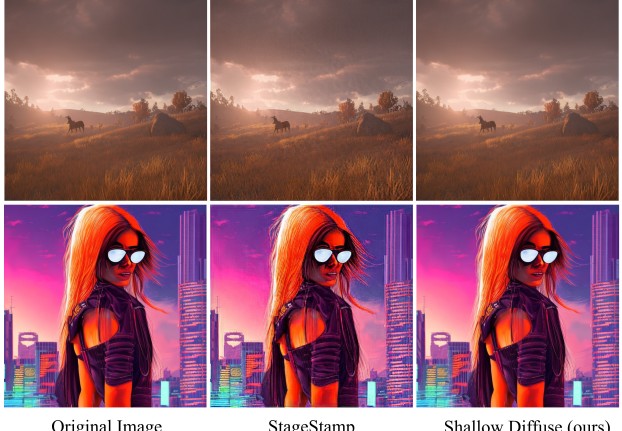

Original Image     StageStamp     Shallow Diffuse (ours)

Figure 7: **Generation Consistency in server scenarios.** We compare the visualization quality of our method against the original image and StageStamp.

The metric for this task is the average robustness across all keys, measured in terms of AUC and TPR@1%FPR. For this study, we test the Tree-Ring and Shallow Diffuse in the server scenario. The results of this experiment are presented in Table 7. Shallow Diffuse consistently outperformed Tree-Ring in robustness across different numbers of users. Even as the number of users increased to 32, Shallow Diffuse maintained strong robustness under clean conditions. However, in adversarial settings, its robustness began to decline when the number of users exceeded 16. Under the current setup, when the number of users surpasses the predefined limit, our method becomes less robust and accurate. We believe that enabling watermarking for hundreds or even thousands of users simultaneously is a challenging yet promising future direction for Shallow Diffuse.

Table 7: **Multi-key re-watermark for different attacks under the server scenario.**

| Watermark numbder | Method | Watermarking Robustness (AUC ↑/TPR@1%FPR↑) | | | | | | | | | | | | |
|---|---|---|---|---|---|---|---|---|---|---|---|---|---|---|
| | | Clean | JPEG | G.Blur | G.Noise | CJ | RR | RD | M.Blur | DiffPure | IC1 | IC2 | IR | Average |
| 2 | Tree-Ring | **1.00/1.00** | 0.99/0.84 | 1.00/0.97 | 0.95/0.83 | 0.98/0.75 | 1.00/1.00 | 1.00/1.00 | 1.00/1.00 | 0.91/0.23 | 1.00/0.91 | 0.98/0.82 | 0.94/0.49 | 0.98/0.80 |
| | Shallow Diffuse | **1.00/1.00** | **1.00/1.00** | **1.00/1.00** | **0.98/0.95** | **1.00/0.90** | **1.00/1.00** | **1.00/1.00** | **1.00/1.00** | **0.98/0.65** | **1.00/0.91** | **1.00/0.97** | **1.00/0.99** | **0.99/0.95** |
| 4 | Tree-Ring | **1.00/1.00** | 0.98/0.63 | 1.00/0.89 | 0.96/0.86 | 0.90/0.54 | 1.00/0.92 | 1.00/0.99 | 1.00/0.95 | 0.88/0.11 | 0.99/0.72 | 0.97/0.67 | 0.92/0.37 | 0.96/0.70 |
| | Shallow Diffuse | **1.00/1.00** | 1.00/0.96 | 0.99/0.88 | 0.97/0.91 | 0.99/0.82 | **1.00/1.00** | **1.00/1.00** | **1.00/1.00** | 0.94/0.37 | 0.99/0.80 | 0.99/0.83 | 0.99/0.89 | 0.99/0.86 |
| 8 | Tree-Ring | 1.00/0.95 | 0.90/0.32 | 0.97/0.56 | 0.92/0.64 | 0.90/0.45 | 0.98/0.71 | 1.00/0.89 | 0.98/0.68 | 0.77/0.08 | 0.91/0.38 | 0.89/0.25 | 0.83/0.16 | 0.91/0.47 |
| | Shallow Diffuse | **1.00/1.00** | 0.99/0.85 | 0.97/0.73 | 0.97/0.90 | 0.98/0.80 | 1.00/0.98 | **1.00/1.00** | 1.00/0.96 | 0.91/0.36 | 0.98/0.71 | 0.97/0.70 | 0.99/0.80 | 0.98/0.80 |
| 16 | Tree-Ring | 0.96/0.57 | 0.78/0.18 | 0.87/0.32 | 0.87/0.38 | 0.84/0.24 | 0.90/0.42 | 0.95/0.53 | 0.90/0.36 | 0.68/0.05 | 0.80/0.18 | 0.77/0.14 | 0.72/0.05 | 0.83/0.26 |
| | Shallow Diffuse | 1.00/0.89 | 0.94/0.59 | 0.89/0.39 | 0.94/0.73 | 0.92/0.53 | 0.97/0.73 | 0.99/0.84 | 0.96/0.73 | 0.78/0.11 | 0.90/0.46 | 0.91/0.46 | 0.92/0.55 | 0.92/0.56 |
| 32 | Tree-Ring | 0.95/0.44 | 0.77/0.11 | 0.85/0.15 | 0.86/0.31 | 0.80/0.15 | 0.88/0.22 | 0.94/0.34 | 0.89/0.26 | 0.63/0.03 | 0.78/0.11 | 0.75/0.08 | 0.70/0.05 | 0.80/0.16 |
| | Shallow Diffuse | 0.99/0.89 | 0.91/0.46 | 0.86/0.26 | 0.93/0.63 | 0.91/0.47 | 0.96/0.65 | 0.99/0.84 | 0.95/0.59 | 0.74/0.07 | 0.87/0.31 | 0.87/0.30 | 0.89/0.28 | 0.90/0.44 |

## C.3 Ablation Study of Different Watermark Patterns

In Table 8, we examine various combinations of watermark patterns $M \odot W$. For the shape of the mask $M$, "Circle" refers to a circular mask $M$ (see Figure 3 top left), while "Ring" represents a ring-shaped $M$. Since the mask is centered in the middle of the figure, "Low" and "High" denote frequency regions: "Low" represents a DFT with zero-frequency centering, whereas "High" indicates a DFT without zero-frequency centering, as illustrated in Figure 3 bottom. For the distribution of $W$, "Zero" implies all values are zero, "Rand" denotes values sampled from $\mathcal{N}(\mathbf{0}, \mathbf{1})$, and "Rotational Rand" represents multiple concentric rings in $W$, with each ring's values sampled from $\mathcal{N}(\mathbf{0}, \mathbf{1})$.

As shown in Table 8, watermarking in high-frequency regions (Rows 7-9) yields improved image consistency compared to low-frequency regions (Rows 1-6). Additionally, the "Circle" $M$ combined with "Rotational Rand" $W$ (Rows 3 and 9) demonstrates greater robustness than other watermark patterns. Consequently, Shallow Diffuse employs the "Circle" $M$ with "Rotational Rand" $W$ in the high-frequency region.

Table 8: **Ablation study on different watermark patterns.**

| Method & Dataset | | | PSNR ↑ | SSIM ↑ | LPIPS ↓ | Average Watermarking Robustness (AUC ↑/TPR@1%FPR↑) |
|---|---|---|---|---|---|---|
| Frequency Region | Shape | Distribution | | | | |
| Low | Circle | Zero | 29.10 | 0.90 | 0.06 | 0.93/0.65 |
| Low | Circle | Rand | 29.37 | 0.92 | 0.05 | 0.92/0.25 |
| Low | Circle | Rotational Rand | 29.13 | 0.90 | 0.06 | **1.00/1.00** |
| Low | Ring | Zero | 36.20 | 0.95 | 0.02 | 0.78/0.35 |
| Low | Ring | Rand | 38.23 | 0.97 | 0.01 | 0.87/0.49 |
| Low | Ring | Rotational Rand | 35.23 | 0.93 | 0.02 | 0.99/0.98 |
| High | Circle | Zero | 38.3 | 0.96 | 0.01 | 0.80/0.34 |
| High | Circle | Rand | **42.3** | **0.98** | **0.004** | 0.86/0.35 |
| High | Circle | Rotational Rand | 38.0 | 0.94 | 0.01 | **1.00/1.00** |

## C.4 Ablation Study of Watermarking Embedded Channel.

As shown in Table 9, we evaluate specific embedding channels $c$ for Shallow Diffuse, where "0," "1," "2," and "3" denote $c = 0, 1, 2, 3$, respectively, and "0 + 1 + 2 + 3" indicates watermarking applied across all channels [4]. Since applying watermarking to any single channel yields similar results (Row 1-4), but applying it to all channels (Row 5) negatively impacts image consistency and robustness, we set $c = 3$ for Shallow Diffuse. The reason is that many image processing operations tend to affect all channels uniformly, making watermarking across all channels more susceptible to such attacks.)

Table 9: **Ablation study on watermarking embedded channel.**

| Watermark embedding channel | PSNR ↑ | SSIM ↑ | LPIPS ↓ | Watermarking Robustness (TPR@1%FPR↑) | | | | |
|---|---|---|---|---|---|---|---|---|
| | | | | Clean | JPEG | G.Blur | G.Noise | Color Jitter |
| 0 | 36.46 | **0.93** | **0.02** | **1.00** | **1.00** | **1.00** | **1.00** | 0.99 |
| 1 | 36.57 | **0.93** | **0.02** | **1.00** | **1.00** | **1.00** | **1.00** | 0.99 |
| 2 | 36.13 | 0.92 | **0.02** | **1.00** | **1.00** | **1.00** | **1.00** | **1.00** |
| 3 | **36.64** | **0.93** | **0.02** | **1.00** | **1.00** | **1.00** | **1.00** | **1.00** |
| 0 + 1 + 2 + 3 | 33.19 | 0.83 | 0.05 | **1.00** | **1.00** | **1.00** | **1.00** | 0.95 |

## C.5 Ablation Study of Inference Steps

We conducted ablation studies on the number of sampling steps, across 10, 25, and 50 steps. The results, shown in Table 10, indicate that Shallow Diffuse is not highly sensitive to sampling steps. The watermark robustness remains consistent across all sampling steps.

Table 10: **Ablation study over inference steps.**

| Steps | Watermarking Robustness (AUC ↑/TPR@1%FPR↑) | | | | | | | | | | | | | |
|---|---|---|---|---|---|---|---|---|---|---|---|---|---|---|
| | Clean | G.Noise | CJ | RD | M.Blur | DiffPure | IC1 | IC2 | DiffDeeper | Rinse2x | Rinse4x | BA | GA | Average |
| 10 | **1.00/1.00** | 0.99/0.89 | 0.95/0.76 | **1.00/1.00** | **1.00/1.00** | **1.00/1.00** | 0.99/0.93 | 0.99/0.93 | 1.00/0.99 | **1.00/1.00** | **1.00/0.98** | **0.63/0.49** | **0.74/0.70** | 0.95/0.90 |
| 25 | **1.00/1.00** | 1.00/0.97 | **1.00/1.00** | **1.00/1.00** | **1.00/1.00** | **1.00/1.00** | 0.99/0.91 | 0.99/0.92 | **1.00/1.00** | **1.00/1.00** | 1.00/0.92 | 0.56/0.48 | 0.73/0.65 | 0.94/0.91 |
| 50 | **1.00/1.00** | **1.00/1.00** | **1.00/1.00** | **1.00/1.00** | **1.00/1.00** | **1.00/1.00** | **1.00/1.00** | **1.00/1.00** | 1.00/0.99 | **1.00/1.00** | 0.99/0.90 | 0.57/0.45 | 0.70/0.63 | **0.94/0.92** |

# D Robustness Analysis on Geometric Distortions

To further analyze robustness under geometric transformations, we conducted an extended study focusing on rotation, cropping, and scaling.

**Controllable Trade-off via Mask Radius.** Our framework enables explicit control over the balance between perceptual quality and geometric robustness by adjusting the frequency mask radius $r$. We compared the original configuration (optimized for visual fidelity) to a more robust configuration with an expanded radius ($r = 3$–13). Results in Table 11 show that increasing the radius improves geometric robustness—particularly rotation and cropping—while incurring only a mild degradation in image consistency.

---

[4]Here we apply Shallow Diffuse on the latent space of Stable Diffusion, the channel dimension is 4.

Table 11: **Trade-off between image fidelity and geometric robustness.** Increasing the mask radius ($r=3$–13) enhances rotation and cropping robustness with minor PSNR drop.

| Setting | CLIP | PSNR ↑ | SSIM ↑ | LPIPS ↓ | Watermarking Robustness (AUC ↑) | | |
| --- | --- | --- | --- | --- | --- | --- | --- |
| | | | | | Rotation | Cropping | Scaling |
| Current ($r=0$–10) | 0.3669 | 35.49 | 0.96 | 0.02 | 0.68 | 0.56 | 1.00 |
| Robust ($r=3$–13) | 0.3637 | 32.05 | 0.95 | 0.03 | 0.90 | 0.89 | 1.00 |

# E  Generalization to Transformer-Based Diffusion Models

To assess whether Shallow Diffuse generalizes beyond U-Net based diffusion architectures, we conducted an additional study on FLUX [73], a transformer-based diffusion model that employs a Flow Matching noise scheduler.

**Experimental Setup.**   All evaluations were performed under a server-side watermarking scenario at $512 \times 512$ resolution. The same watermark design as in the Stable Diffusion experiments was used, with two key modifications to account for architectural differences: (a) the watermark radius was set to 5, and (b) watermark injection was applied across all latent channels.

We generated 100 watermarked images and evaluated both consistency and robustness across injection timesteps $\{0.1T, 0.2T, \ldots, 0.9T\}$. The results are presented in Table 12.

Table 12: **Generalization of Shallow Diffuse to the FLUX transformer-based diffusion model.**

| Timestep ($t/T$) | PSNR ↑ | SSIM ↑ | LPIPS ↓ | Watermarking Robustness | | |
| --- | --- | --- | --- | --- | --- | --- |
| | | | | AUC ↑ | ACC ↑ | TPR@1%FPR ↑ |
| 0.1 | 31.27 | 0.91 | 0.05 | 0.90 | 0.93 | **0.87** |
| 0.2 | 31.66 | 0.92 | **0.04** | 0.91 | 0.93 | **0.87** |
| 0.3 | 31.68 | 0.92 | 0.05 | 0.92 | **0.94** | **0.87** |
| 0.4 | **31.69** | **0.93** | **0.04** | 0.93 | **0.94** | 0.86 |
| 0.5 | 31.68 | **0.93** | **0.04** | **0.94** | 0.93 | 0.86 |
| 0.6 | 31.56 | **0.93** | **0.04** | **0.94** | **0.94** | 0.85 |
| 0.7 | 31.50 | **0.93** | **0.04** | **0.94** | 0.93 | 0.81 |
| 0.8 | 31.52 | **0.93** | 0.05 | **0.94** | **0.94** | 0.82 |
| 0.9 | 31.62 | 0.92 | 0.06 | **0.94** | **0.94** | 0.81 |

**Analysis.**   To ensure a fair comparison of injection timesteps across different schedulers, we matched the effective Signal-to-Noise Ratio (SNR) between the Variance Preserving (VP) schedule used in Stable Diffusion and the Flow Matching scheduler in FLUX. A timestep of $t/T = 0.3$ in VP approximately corresponds to $t/T = 0.205$ in Flow Matching when equalizing SNR.

The results indicate that injecting at this equivalent "shallow" timestep achieves the best SSIM (0.93) and near-optimal PSNR (31.7), while also maximizing robustness (AUC 0.94, TPR@1%FPR 0.87). This confirms that the optimal embedding region discovered for Stable Diffusion generalizes to transformer-based architectures, underscoring the broad applicability of our null-space embedding framework.

# F  Comparison with ROBIN

We conduct a direct empirical comparison between our optimization-free Shallow Diffuse and the optimization-based ROBIN [32] under the server scenario with 1000 generations. Experiment results are shown in Table 13 and Table 14.

Our experiments show that Shallow Diffuse produces images with significantly higher perceptual quality and consistency. As shown in Table 13, our method achieves a PSNR nearly 11 dB higher and a better FID score. We attribute the difference to the frequency domains where watermarks are added: Shallow Diffuse uses the high-frequency domain, while ROBIN uses the low-frequency domain.

Adding the watermark to high frequencies preserves the low-frequency content of the generated image, thereby significantly improving consistency and quality.

In terms of robustness, the two methods are competitive, each with distinct strengths. Table 14 shows that both methods are highly robust to most distortion and regeneration attacks. ROBIN demonstrates superior robustness against geometric attacks like rotation (1.0 vs 0.69) and cropping (0.99 vs 0.58), as well as adversarial attacks.

This empirical comparison quantifies the fundamental trade-off. ROBIN's optimization process achieves higher robustness for challenging attacks at the cost of significantly lower image quality, longer setup times, and less flexibility. Shallow Diffuse provides a more balanced and practical solution, offering state-of-the-art image quality and comparable robustness across a wide range of common attacks, all within an efficient, optimization-free framework adaptable to both server and user needs.

Table 13: **Consistency between Shallow Diffuse and ROBIN under the server scenario.**

| Method | PSNR $\uparrow$ | SSIM $\uparrow$ | LPIPS $\downarrow$ | FID $\downarrow$ | CLIP $\uparrow$ |
|---|---|---|---|---|---|
| ROBIN | 24.614 | 0.8261 | 0.1087 | 134.8 | 0.366 |
| Shallow Diffuse | **35.49** | **0.96** | **0.05** | **129.228** | **0.367** |

Table 14: **Robustness between Shallow Diffuse and ROBIN under the server scenario.**

| Method | Watermarking Robustness (AUC) $\uparrow$ | | | | | | | | | | | | | | |
|---|---|---|---|---|---|---|---|---|---|---|---|---|---|---|---|
| | JPEG | G.Blur | G.Noise | CJ | RR | RD | M.Blur | Rotation | Crop | IC1 | IC2 | IR | Rinse4x | BA | GA |
| ROBIN | **0.999** | **0.999** | 0.963 | 0.962 | **1** | **1** | **1** | **1** | **0.991** | 0.998 | **1** | **1** | **1** | **0.939** | **0.9** |
| Shallow Diffuse | **0.999** | **0.999** | **0.997** | **0.967** | **1** | **1** | **1** | 0.691 | 0.582 | **1** | 0.998 | 0.993 | 0.999 | 0.67 | 0.78 |

# G  Proofs in Section 4

## G.1  Proofs of Theorem 1

*Proof of Theorem 1.* According to Assumption 1, we have $||\hat{\boldsymbol{x}}_{0,t}^{\mathcal{W}} - \hat{\boldsymbol{x}}_{0,t}||_2^2 = \lambda||\boldsymbol{J}_{\boldsymbol{\theta},t}(\boldsymbol{x}_t) \cdot \Delta\boldsymbol{x}||_2^2$. From Levy's Lemma proposed in [74], given function $||\boldsymbol{J}_{\boldsymbol{\theta},t}(\boldsymbol{x}_t) \cdot \Delta\boldsymbol{x}||_2^2 : \mathbb{S}^{d-1} \to \mathbb{R}$ we have:

$$\mathbb{P}\left(\left|||\boldsymbol{J}_{\boldsymbol{\theta},t}(\boldsymbol{x}_t) \cdot \Delta\boldsymbol{x}||_2^2 - \mathbb{E}\left[||\boldsymbol{J}_{\boldsymbol{\theta},t}(\boldsymbol{x}_t) \cdot \Delta\boldsymbol{x}||_2^2\right]\right| \geq \epsilon\right) \leq 2\exp\left(\frac{-C(d-2)\epsilon^2}{L^2}\right),$$

given $L$ to be the Lipschitz constant of $||\boldsymbol{J}_{\boldsymbol{\theta},t}(\boldsymbol{x}_t)||_2^2$ and $C$ is a positive constant (which can be taken to be $C = (18\pi^3)^{-1}$). From Lemma 2 and Lemma 3, we have:

$$\mathbb{P}\left(\left|||\boldsymbol{J}_{\boldsymbol{\theta},t}(\boldsymbol{x}_t) \cdot \Delta\boldsymbol{x}||_2^2 - \frac{||\boldsymbol{J}_{\boldsymbol{\theta},t}(\boldsymbol{x}_t)||_F^2}{d}\right| \geq \epsilon\right) \leq 2\exp\left(\frac{-(18\pi^3)^{-1}(d-2)\epsilon^2}{||\boldsymbol{J}_{\boldsymbol{\theta},t}(\boldsymbol{x}_t)||_2^4}\right).$$

Define $\dfrac{1}{r_t}$ as the desired probability level, set

$$\frac{1}{r_t} = 2\exp\left(\frac{-(18\pi^3)^{-1}(d-2)\epsilon^2}{||\boldsymbol{J}_{\boldsymbol{\theta},t}(\boldsymbol{x}_t)||_2^4}\right),$$

Solving for $\epsilon$:

$$\epsilon = ||\boldsymbol{J}_{\boldsymbol{\theta},t}(\boldsymbol{x}_t)||_2^2\sqrt{\frac{18\pi^3}{d-2}\log(2r_t)}.$$

Therefore, with probability $1 - \dfrac{1}{r_t}$, we have:

$$||\hat{\boldsymbol{x}}_{0,t}^{\mathcal{W}} - \hat{\boldsymbol{x}}_{0,t}||_2^2 = \lambda^2 ||\boldsymbol{J}_{\boldsymbol{\theta},t}(\boldsymbol{x}_t) \cdot \Delta \boldsymbol{x}||_2^2,$$

$$\leq \frac{\lambda^2 ||\boldsymbol{J}_{\boldsymbol{\theta},t}(\boldsymbol{x}_t)||_F^2}{d} + \lambda^2 ||\boldsymbol{J}_{\boldsymbol{\theta},t}(\boldsymbol{x}_t)||_2^2 \sqrt{\frac{18\pi^3}{d-2} \log{(2r_t)}},$$

$$\leq \lambda^2 ||\boldsymbol{J}_{\boldsymbol{\theta},t}(\boldsymbol{x}_t)||_2^2 \left( \frac{r_t}{d} + \sqrt{\frac{18\pi^3}{d-2} \log{(2r_t)}} \right),$$

$$= \lambda^2 L^2 \left( \frac{r_t}{d} + \sqrt{\frac{18\pi^3}{d-2} \log{(2r_t)}} \right),$$

where the last inequality is obtained from $||\boldsymbol{J}_{\boldsymbol{\theta},t}(\boldsymbol{x}_t)||_F^2 \leq r_t ||\boldsymbol{J}_{\boldsymbol{\theta},t}(\boldsymbol{x}_t)||_2^2$. Therefore, with probability $1 - \dfrac{1}{r_t}$,

$$||\hat{\boldsymbol{x}}_{0,t}^{\mathcal{W}} - \hat{\boldsymbol{x}}_{0,t}||_2 \leq \lambda L \sqrt{\frac{r_t}{d} + \sqrt{\frac{18\pi^3}{d-2} \log{(2r_t)}}} = \lambda L h(r_t).$$

$\square$

*Proof of Theorem 2.* According to Equation (1), one step of DDIM sampling at timestep $t$ could be represented by PMP $\boldsymbol{f}_{\boldsymbol{\theta},t}(\boldsymbol{x}_t)$ as:

$$\boldsymbol{x}_{t-1} = \sqrt{\alpha_{t-1}} \boldsymbol{f}_{\boldsymbol{\theta},t}(\boldsymbol{x}_t) + \sqrt{1-\alpha_{t-1}} \left( \frac{\boldsymbol{x}_t - \sqrt{\alpha_t}\boldsymbol{f}_{\boldsymbol{\theta},t}(\boldsymbol{x}_t)}{\sqrt{1-\alpha_t}} \right), \tag{10}$$

$$= \sqrt{\frac{1-\alpha_{t-1}}{1-\alpha_t}} \boldsymbol{x}_t + \frac{\sqrt{1-\alpha_t}\sqrt{\alpha_{t-1}} - \sqrt{1-\alpha_{t-1}}\sqrt{\alpha_t}}{\sqrt{1-\alpha_t}} \boldsymbol{f}_{\boldsymbol{\theta},t}(\boldsymbol{x}_t), \tag{11}$$

If we inject a watermark $\lambda\Delta\boldsymbol{x}$ to $\boldsymbol{x}_t$, so $x_t^{\mathcal{W}} = \boldsymbol{x}_t + \lambda\Delta\boldsymbol{x}$. To solve $x_{t-1}^{\mathcal{W}}$, we could plugging Equation (2) to Equation (11), we could obtain:

$$\boldsymbol{x}_{t-1}^{\mathcal{W}} = \sqrt{\frac{1-\alpha_{t-1}}{1-\alpha_t}} \boldsymbol{x}_t^{\mathcal{W}} + \frac{\sqrt{1-\alpha_t}\sqrt{\alpha_{t-1}} - \sqrt{1-\alpha_{t-1}}\sqrt{\alpha_t}}{\sqrt{1-\alpha_t}} \boldsymbol{f}_{\boldsymbol{\theta},t}(\boldsymbol{x}_t^{\mathcal{W}}), \tag{12}$$

$$= \boldsymbol{x}_{t-1} + \sqrt{\frac{1-\alpha_{t-1}}{1-\alpha_t}} \lambda\Delta\boldsymbol{x} + \frac{\sqrt{1-\alpha_t}\sqrt{\alpha_{t-1}} - \sqrt{1-\alpha_{t-1}}\sqrt{\alpha_t}}{\sqrt{1-\alpha_t}} \boldsymbol{J}_{\boldsymbol{\theta},t}(\boldsymbol{x}_t)\Delta\boldsymbol{x} \tag{13}$$

$$= \boldsymbol{x}_{t-1} + \lambda \underbrace{\left( \sqrt{\frac{1-\alpha_{t-1}}{1-\alpha_t}} \boldsymbol{I} + \frac{\sqrt{1-\alpha_t}\sqrt{\alpha_{t-1}} - \sqrt{1-\alpha_{t-1}}\sqrt{\alpha_t}}{\sqrt{1-\alpha_t}} \boldsymbol{J}_{\boldsymbol{\theta},t}(\boldsymbol{x}_t) \right)}_{:=\boldsymbol{W}_t} \Delta\boldsymbol{x}, \tag{14}$$

One step DDIM Inverse sampling at timestep $t-1$ could be represented by PMP $\boldsymbol{f}_{\boldsymbol{\theta},t}(\boldsymbol{x}_t)$ as:

$$\boldsymbol{x}_t = \sqrt{\frac{1-\alpha_t}{1-\alpha_{t-1}}} \boldsymbol{x}_{t-1} + \frac{\sqrt{1-\alpha_{t-1}}\sqrt{\alpha_t} - \sqrt{1-\alpha_t}\sqrt{\alpha_{t-1}}}{\sqrt{1-\alpha_{t-1}}} \boldsymbol{f}_{\boldsymbol{\theta},t-1}(\boldsymbol{x}_{t-1}), \tag{15}$$

To detect the watermark, we apply one step DDIM Inverse on $\boldsymbol{x}_{t-1}^{\mathcal{W}}$ at timestep $t-1$ to obtain $\tilde{x}_t^{\mathcal{W}}$:

$$\tilde{x}_t^{\mathcal{W}} = \sqrt{\frac{1-\alpha_t}{1-\alpha_{t-1}}} \boldsymbol{x}_{t-1}^{\mathcal{W}} + \frac{\sqrt{1-\alpha_{t-1}}\sqrt{\alpha_t} - \sqrt{1-\alpha_t}\sqrt{\alpha_{t-1}}}{\sqrt{1-\alpha_{t-1}}} \boldsymbol{f}_{\boldsymbol{\theta},t-1}(\boldsymbol{x}_{t-1}^{\mathcal{W}}),$$

$$= \boldsymbol{x}_t + \lambda \underbrace{\left( \sqrt{\frac{1-\alpha_t}{1-\alpha_{t-1}}} \boldsymbol{I} + \frac{\sqrt{1-\alpha_{t-1}}\sqrt{\alpha_t} - \sqrt{1-\alpha_t}\sqrt{\alpha_{t-1}}}{\sqrt{1-\alpha_{t-1}}} \boldsymbol{J}_{\boldsymbol{\theta},t-1}(\boldsymbol{x}_{t-1}) \right)}_{:=\boldsymbol{W}_{t-1}} \boldsymbol{W}_t\Delta\boldsymbol{x},$$

$$= \boldsymbol{x}_t + \lambda\boldsymbol{W}_{t-1}\boldsymbol{W}_t\Delta\boldsymbol{x} = \boldsymbol{x}_t^{\mathcal{W}} + \lambda\left(\boldsymbol{W}_{t-1}\boldsymbol{W}_t - \boldsymbol{I}\right)\Delta\boldsymbol{x}.$$

Therefore:

$$||\tilde{x}_t^{\mathcal{W}} - \boldsymbol{x}_t^{\mathcal{W}}||_2 = \lambda ||\left(\boldsymbol{W}_{t-1}\boldsymbol{W}_t - \boldsymbol{I}\right)\Delta\boldsymbol{x}||_2,$$

$$= \lambda||\frac{\sqrt{1-\alpha_{t-1}}\sqrt{\alpha_t} - \sqrt{1-\alpha_t}\sqrt{\alpha_{t-1}}}{\sqrt{1-\alpha_t}}\boldsymbol{J}_{\boldsymbol{\theta},t-1}(\boldsymbol{x}_{t-1})\Delta\boldsymbol{x},$$

$$+ \frac{\sqrt{1-\alpha_t}\sqrt{\alpha_{t-1}} - \sqrt{1-\alpha_{t-1}}\sqrt{\alpha_t}}{\sqrt{1-\alpha_{t-1}}}\boldsymbol{J}_{\boldsymbol{\theta},t}(\boldsymbol{x}_t)\Delta\boldsymbol{x},$$

$$- \frac{\left(\sqrt{1-\alpha_t}\sqrt{\alpha_{t-1}} - \sqrt{1-\alpha_{t-1}}\sqrt{\alpha_t}\right)^2}{\sqrt{1-\alpha_{t-1}}\sqrt{1-\alpha_t}}\boldsymbol{J}_{\boldsymbol{\theta},t-1}(\boldsymbol{x}_{t-1})\boldsymbol{J}_{\boldsymbol{\theta},t}(\boldsymbol{x}_t)\Delta\boldsymbol{x}||_2,$$

$$\leq -\lambda g\left(\alpha_t,\alpha_{t-1}\right)||\boldsymbol{J}_{\boldsymbol{\theta},t-1}(\boldsymbol{x}_{t-1})\Delta\boldsymbol{x}||_2 + \lambda g\left(\alpha_{t-1},\alpha_t\right)||\boldsymbol{J}_{\boldsymbol{\theta},t}(\boldsymbol{x}_t)\Delta\boldsymbol{x}||_2$$

$$- \lambda g\left(\alpha_{t-1},\alpha_t\right)g\left(\alpha_t,\alpha_{t-1}\right)||\boldsymbol{J}_{\boldsymbol{\theta},t-1}(\boldsymbol{x}_{t-1})\boldsymbol{J}_{\boldsymbol{\theta},t}(\boldsymbol{x}_t)\Delta\boldsymbol{x}||_2,$$

$$\leq -\lambda g\left(\alpha_t,\alpha_{t-1}\right)||\boldsymbol{J}_{\boldsymbol{\theta},t-1}(\boldsymbol{x}_{t-1})\Delta\boldsymbol{x}||_2$$

$$+ \lambda g\left(\alpha_{t-1},\alpha_t\right)\left(1 - g\left(\alpha_t,\alpha_{t-1}\right)L\right)||\boldsymbol{J}_{\boldsymbol{\theta},t}(\boldsymbol{x}_t)\Delta\boldsymbol{x}||_2,$$

$$= -g\left(\alpha_t,\alpha_{t-1}\right)||\hat{\boldsymbol{x}}_{0,t-1}^{\mathcal{W}} - \hat{\boldsymbol{x}}_{0,t-1}||_2$$

$$+ g\left(\alpha_{t-1},\alpha_t\right)\left(1 - g\left(\alpha_t,\alpha_{t-1}\right)L\right)||\hat{\boldsymbol{x}}_{0,t}^{\mathcal{W}} - \hat{\boldsymbol{x}}_{0,t}||_2,$$

The first inequality holds because $g\left(\alpha_{t-1},\alpha_t\right) < 0$ and $g\left(\alpha_t,\alpha_{t-1}\right) > 0$. The second inequality holds because $||\boldsymbol{J}_{\boldsymbol{\theta},t-1}(\boldsymbol{x}_{t-1})\boldsymbol{J}_{\boldsymbol{\theta},t}(\boldsymbol{x}_t)\Delta\boldsymbol{x}||_2 \leq ||\boldsymbol{J}_{\boldsymbol{\theta},t-1}(\boldsymbol{x}_{t-1})||_2 ||\boldsymbol{J}_{\boldsymbol{\theta},t}(\boldsymbol{x}_t)\Delta\boldsymbol{x}||_2 \leq L||\boldsymbol{J}_{\boldsymbol{\theta},t}(\boldsymbol{x}_t)\Delta\boldsymbol{x}||_2$. From Theorem 1, with probability $1 - \frac{1}{r_{t-1}}$,

$$||\hat{\boldsymbol{x}}_{0,t-1}^{\mathcal{W}} - \hat{\boldsymbol{x}}_{0,t-1}||_2 \leq \lambda L h(r_{t-1}),$$

with probability $1 - \frac{1}{r_t}$,

$$||\hat{\boldsymbol{x}}_{0,t}^{\mathcal{W}} - \hat{\boldsymbol{x}}_{0,t}||_2 \leq \lambda L h(r_t),$$

Thus, from the union of bound, with a probability at least $1 - \frac{1}{r_t} - \frac{1}{r_{t-1}}$,

$$||\tilde{x}_t^{\mathcal{W}} - \boldsymbol{x}_t^{\mathcal{W}}||_2 \leq -\lambda L g\left(\alpha_t,\alpha_{t-1}\right)h(r_{t-1}) + \lambda L g\left(\alpha_{t-1},\alpha_t\right)\left(1 - g\left(\alpha_t,\alpha_{t-1}\right)L\right)h(r_t)$$

$$\leq \lambda L\left(-g\left(\alpha_t,\alpha_{t-1}\right) + g\left(\alpha_{t-1},\alpha_t\right)\left(1 - Lg\left(\alpha_t,\alpha_{t-1}\right)\right)\right)h(\max\{r_{t-1},r_t\})$$

$\square$

## H Auxiliary Results

**Lemma 1.** *Given a unit vector $\boldsymbol{v}_i$ with and $\boldsymbol{\epsilon} \sim \mathcal{N}(\boldsymbol{0},\boldsymbol{I}_d)$, we have*

$$\mathbb{E}_{\boldsymbol{\epsilon}\sim\mathcal{N}(\boldsymbol{0},\boldsymbol{I}_d)}[\left(\boldsymbol{v}_i^T\boldsymbol{\epsilon}\right)^2/||\boldsymbol{\epsilon}||_2^2] = \frac{1}{d}.$$

*Proof of Lemma 1.* Because $\boldsymbol{\epsilon} \sim \mathcal{N}(\boldsymbol{0},\boldsymbol{I}_d)$,

$$\boldsymbol{v}_i^T\boldsymbol{\epsilon} \sim \mathcal{N}(\boldsymbol{v}_i^T\boldsymbol{0},\boldsymbol{v}_i^T\boldsymbol{I}_d\boldsymbol{v}_i) = \mathcal{N}(\boldsymbol{v}_i^T\boldsymbol{0},\boldsymbol{v}_i^T\boldsymbol{I}_d\boldsymbol{v}_i) = \mathcal{N}(0,1), \tag{16}$$

Assume a set of $d$ unit vecotrs $\{v_1,v_2,\ldots,\boldsymbol{v}_i,\ldots,v_d\}$ are orthogonormal and are basis of $\mathbb{R}^d$, similarly, we could show that $\forall j \in [d], X_j := v_j^T\boldsymbol{\epsilon} \sim \mathcal{N}(0,1)$. Therefore, we could rewrite $\left(\boldsymbol{v}_i^T\boldsymbol{\epsilon}\right)^2/||\boldsymbol{\epsilon}||_2^2$ as:

$$\left(\boldsymbol{v}_i^T\boldsymbol{\epsilon}\right)^2/||\boldsymbol{\epsilon}||_2^2 = \frac{\left(\boldsymbol{v}_i^T\boldsymbol{\epsilon}\right)^2}{||\sum_{k=1}^d v_k v_k^T\boldsymbol{\epsilon}||_2^2}, \tag{17}$$

$$= \frac{\left(\boldsymbol{v}_i^T\boldsymbol{\epsilon}\right)^2}{\sum_{k=1}^d \left(v_k^T\boldsymbol{\epsilon}\right)^2}, \tag{18}$$

$$= \frac{X_i^2}{\sum_{k=1}^d X_k^2}. \tag{19}$$

Let $Y_i := \frac{X_i^2}{\sum_{j=1}^d X_j^2}$. Because $\forall j \in [d], X_j := v_j^T \epsilon \sim \mathcal{N}(0,1), \forall j \in [d], Y_j$ has the same distribution. Additionally, $\sum_{j=1}^d Y_j = 1$. So:

$$\mathbb{E}_{\epsilon \sim \mathcal{N}(\mathbf{0}, \mathbf{I}_d)}[\frac{\left(v_i^T \epsilon\right)^2}{||\epsilon||_2^2}] = \mathbb{E}[Y_i] = \frac{1}{d}\mathbb{E}[\sum_{j=1}^d Y_j] = \frac{1}{d}.$$

$\square$

**Lemma 2.** *Given a matrix $\mathbf{J} \in \mathbb{R}^{d \times d}$ with $\mathrm{rank}(\mathbf{J}) = r$. Given $\boldsymbol{x}$ which is uniformly sampled on the unit hypersphere $\mathbb{S}^{d-1}$, we have:*

$$\mathbb{E}_{\boldsymbol{x}}\left[||\mathbf{J}\boldsymbol{x}||_2^2\right] = \frac{||\mathbf{J}||_F^2}{d}.$$

*Proof of Lemma 2.* Let's define the singular value decomposition of $\mathbf{J} = \mathbf{U}\boldsymbol{\Sigma}\mathbf{V}^T$ with $\boldsymbol{\Sigma} = \mathrm{diag}(\sigma_1, \ldots, \sigma_r, 0 \ldots, 0)$. Therefore, $\mathbb{E}_{\boldsymbol{x}}\left[||\mathbf{J}\boldsymbol{x}||_2^2\right] = \mathbb{E}_{\boldsymbol{x}}\left[||\mathbf{U}\boldsymbol{\Sigma}\mathbf{V}^T\boldsymbol{x}||_2^2\right] = \mathbb{E}_{\boldsymbol{z}}\left[||\boldsymbol{\Sigma}\boldsymbol{z}||_2^2\right]$ where $\boldsymbol{z} := \mathbf{V}^T\boldsymbol{x}$ is is uniformly sampled on the unit hypersphere $\mathbb{S}^{d-1}$. Thus, we have:

$$\mathbb{E}_{\boldsymbol{z}}\left[||\boldsymbol{\Sigma}\boldsymbol{z}||_2^2\right] = \mathbb{E}_{\boldsymbol{z}}\left[||\sum_{i=1}^r \sigma_i e_i^T \boldsymbol{z}||_2^2\right],$$

$$= \mathbb{E}_{\boldsymbol{z}}\left[\sum_{i=1}^r \sigma_i^2 ||e_i^T \boldsymbol{z}||_2^2\right],$$

$$= \sum_{i=1}^r \sigma_i^2 \mathbb{E}_{\boldsymbol{z}}\left[||e_i^T \boldsymbol{z}||_2^2\right] = \frac{||\mathbf{J}||_F^2}{d},$$

where $e_i$ is the standard basis with $i$-th element equals to 0. The second equality is because of independence between $e_i^T \boldsymbol{z}$ and $e_j^T \boldsymbol{z}$. The fourth equality is from Lemma 1. $\square$

**Lemma 3.** *Given function $f(\boldsymbol{x}) = ||\mathbf{J}\boldsymbol{x}||_2^2$, the lipschitz constant $L_f$ of function $f(\boldsymbol{x})$ is:*

$$L_f = 2||\mathbf{J}||_2^2.$$

*Proof of Lemma 3.* The jacobian of $f(\boldsymbol{x})$ is:

$$\nabla_{\boldsymbol{x}} f(\boldsymbol{x}) = 2\mathbf{J}^T\mathbf{J}\boldsymbol{x},$$

Therefore, the lipschitz constant $L$ follows:

$$L_f = \sup_{\boldsymbol{x} \in \mathbb{S}^{d-1}} ||\nabla_{\boldsymbol{x}} f(\boldsymbol{x})||_2 = 2 \sup_{\boldsymbol{x} \in \mathbb{S}^{d-1}} ||\mathbf{J}^T\mathbf{J}\boldsymbol{x}||_2 = ||\mathbf{J}^T\mathbf{J}||_2 = ||\mathbf{J}||_2^2$$

$\square$

