# OpenReview forum: "Shallow Diffuse: Robust and Invisible Watermarking through Low-Dim Subspaces in Diffusion Models"
_NeurIPS.cc/2025/Conference — NeurIPS 2025 spotlight_

### Official Review · Reviewer_LsBM · 2025-06-29

**Clarity:** 3
**Significance:** 3
**Originality:** 2
**Rating:** 5
**Confidence:** 4

**Summary:**

This work proposes a diffusion model watermarking technique that embeds signals into a low-dimensional subspace’s null space, aiming to decouple watermark information from image content. The method claims to maintain image fidelity while ensuring robustness against attacks, supported by experiments comparing favorably against existing approaches in quality preservation and robustness.

**Questions:**

1. The two primary distinctions from Tree-Ring – (a) later-stage embedding and (b) high-frequency localization – appear impactful. Could the authors provide ablation studies clarifying how each contributes to robustness/image quality?

2. Conventional wisdom suggests low-frequency watermarks resist signal processing (compression/filtering) but suffer poor geometric robustness, while high-frequency variants show the inverse tradeoff. Yet Appendix Table 8 reports **identical robustness (1.00)** for low/high-frequency variants (rotational rand circle) under tested distortions. *Would stronger attacks (e.g., aggressive compression or rotation) reveal expected tradeoffs?*

**Ethical Concerns:**

["NO or VERY MINOR ethics concerns only"]

**Final Justification:**

See the comments.

**Limitations:**

yes

**Paper Formatting Concerns:**

No formatting concerns

**Quality:**

3

**Strengths And Weaknesses:**

### Strengths
1. The focus on authenticating generative model outputs addresses a timely security challenge in real-world deployments.
2. Leveraging low-rank PMP Jacobian properties to minimize watermark interference without model retraining demonstrates analytical insight.
3. The paper is logically structured and technically accessible, with clear motivation and methodology.

### Weaknesses
1. **Robustness Validation:** While multiple attack scenarios are tested, inclusion of **rotation robustness evaluation** would strengthen claims given its prevalence in image manipulations.
2. **Contextual Positioning:** Comparison with Robin [1] —which similarly embeds watermarks at shallow diffusion steps and aims for semantic preservation—would better situate the claimed advantages in semantic preservation.

[1] ROBIN: Robust and Invisible Watermarks for Diffusion Models with Adversarial Optimization. NeurIPS 2024.

---

> ### Author Rebuttal · Authors · 2025-07-31
>
> > W1: Robustness validation: inclusion of rotation robustness evaluation
>
> A1:
>
> We thank the reviewer for this valuable suggestion. We agree that evaluating against rotation is a critical component of a comprehensive robustness assessment.
>
> To empirically validate our design and directly address the reviewer's point, we have conducted new experiments evaluating rotation robustness. We found that there is a controllable trade-off between image consistency and rotation robustness, which can be managed by adjusting the watermark's frequency mask radius.
>
> The table below presents these new results, comparing the original paper's setting (optimized for image quality) with a new setting optimized for geometric robustness. We present the robustness performance in AUC.
>
> | setting               | CLIP   | PSNR  | SSIM | LPIPS | rotation |
> |-----------------------|--------|-------|------|--------|----------|
> | current setting (0-10)| 0.3669 | 35.49 | 0.96 | 0.02   | 0.68     |
> | r=3-13                | 0.3637 | 32.05 | 0.95 | 0.03   | 0.91     |
>
> As the results show, by adjusting the parameters (r=3-13 setting), we can significantly improve robustness against rotation to a score of 0.91. This is achieved with only a minor trade-off in image quality (PSNR remains high at 32.05), which is still superior to other diffusion-based methods presented in our original paper's Table 1.
>
> ---
> > W2: Context positioning: comparison with Robin
>
> A2:
>
> We thank the reviewer for their insightful feedback and for encouraging us to position our work relative to ROBIN [1]. We agree that this comparison is important for highlighting the unique contributions of Shallow Diffuse.
>
> The most crucial distinction is that Shallow Diffuse is an entirely training-free and optimization-free method, whereas ROBIN relies on a computationally intensive adversarial optimization process to achieve its results.
>
> We can break down the key differences as follows:
>
> 1. **Mechanism for Invisibility:**
>
> ROBIN employs an active hiding process. It first embeds a strong watermark and then uses an adversarial optimization algorithm to generate a special "hiding prompt guidance signal" (w_p). This signal actively guides the diffusion model during the later generation steps to conceal the visual artifacts introduced by the watermark. This is a multi-stage process of "implant then hide."
>
> Shallow Diffuse achieves invisibility passively and elegantly. Our method does not require any hiding signal or optimization. Instead, we leverage the intrinsic low-rank property of the diffusion model's Jacobian. By embedding the watermark in the high-dimensional null space of this Jacobian, it is naturally decoupled from the image content. This ensures minimal visual distortion as a direct consequence of our embedding strategy, a concept supported by our theoretical analysis in Theorem 1.
>
> 2. **Methodology and Cost:**
>
> ROBIN requires a dedicated adversarial optimization phase to discover the optimal watermark (w_i) and its corresponding hiding prompt (w_p). This adds a layer of computational overhead before the watermarking can be applied.
>
> Shallow Diffuse has no such requirement. Our method is a direct, one-shot injection process based on pre-defined watermark patterns. This makes it more lightweight, efficient, and practical for real-world deployment where on-the-fly watermarking is often necessary.
>
> 3. **Foundation of Robustness:**
>
> ROBIN's robustness stems from its ability to embed an initially stronger watermark, with the assumption that the subsequent active hiding process will successfully conceal it.
>
> Shallow Diffuse's robustness is grounded in its decoupling principle. The watermark is carried through the DDIM sampling process in a mathematical term separate from the primary image content (as shown in our paper's Equation 5). This design makes the watermark signal theoretically less susceptible to degradation from attacks on the final image's visual features, a property supported by our strong experimental results against a wide range of attacks.
>
> In summary, while both methods insightfully embed watermarks at an intermediate diffusion step, they operate on fundamentally different principles. ROBIN introduces an active hiding mechanism powered by adversarial optimization. In contrast, Shallow Diffuse is a training-free solution that achieves both invisibility and robustness by exploiting the inherent geometric structure of the diffusion model itself.
>
> We believe these distinctions clearly highlight the unique contribution of our work. In our revised manuscript, we will expand our related works section to include a thorough discussion of ROBIN, detailing both the similarities in approach (i.e., embedding at an intermediate step) and the important differences in mechanism (optimization-based active hiding vs. optimization-free passive decoupling). This will better situate our work and clarify its significance. We thank the reviewer for this valuable suggestion.
>
> [1] Huang, H., Wu, Y., & Wang, Q. (2024). Robin: Robust and invisible watermarks for diffusion models with adversarial optimization. Advances in Neural Information Processing Systems, 37, 3937-3963.
>
> ---
>
> > Q1: Provide ablation studies on (a) later-stage embedding and (b) high-frequency localization.
>
> A3:
> We thank the reviewer for this excellent suggestion to clarify the individual contributions of our method's key components. We have conducted a new ablation study to precisely disentangle the effects of (a) later-stage embedding and (b) high-frequency localization. The results, presented in the table below, confirm that both components are crucial and contribute synergistically to the superior performance of Shallow Diffuse.
>
> | watermark method for ablation      | CLIP-Score | FID   | PSNR  | SSIM | LPIPS | Distortion | Regeneration | Adversarial |
> |------------------------------------|------------|-------|-------|------|--------|-------------|---------------|-------------|
> | tree-ring watermark                | 0.3622     | 25.82 | 16.61 | 0.64 | 0.31   | 0.88        | 0.87          | 0.06        |
> | tree-ring high frequency           | 0.3656     | 25.72 | 25.9  | 0.84 | 0.10   | 0.92        | 0.89          | 0.35        |
> | tree-ring later-stage embedding    | 0.3648     | 25.74 | 20.68 | 0.94 | 0.05   | 0.92        | 0.95          | 0.44        |
> | shallow diffuse (ours)             | 0.3669     | 25.6  | 35.49 | 0.96 | 0.02   | 1           | 0.98          | 0.54        |
>
> - **Baseline (Tree-Ring Watermark)**: The first row establishes the baseline performance, replicating the original Tree-Ring method. It shows low image consistency (PSNR of 16.61) and particularly low robustness against adversarial attacks (0.06).
> - **Contribution of High-Frequency Localization**: To isolate this factor, we modified the baseline to embed the watermark in high frequencies while retaining the early-stage embedding (t=1.0T). As shown in the second row, this single change significantly improves image consistency (PSNR jumps from 16.61 to 25.9) and boosts adversarial robustness by over 5x (from 0.06 to 0.35).
> - **Contribution of Later-Stage Embedding**: To isolate this second factor, we used the original low-frequency watermark but injected it at our proposed intermediate timestep (t=0.3T). The third row shows this also improves consistency (PSNR to 20.68) and yields an even greater improvement in adversarial robustness (from 0.06 to 0.44).
> - **Synergistic Effect (Shallow Diffuse)**: Our proposed method, Shallow Diffuse, combines both high-frequency localization and later-stage embedding. The final row demonstrates a clear synergistic effect: it achieves a state-of-the-art PSNR of 35.49 and the highest robustness across all attack categories, outperforming the sum of the individual modifications.
>
> This detailed ablation validates that both of our key technical contributions are impactful on their own and work together to achieve the superior performance of Shallow Diffuse. We will add this table and analysis to the appendix of our future manuscript.
>
> ---
>
> > Q2: Table 8 reports identical robustness for low/high-frequency variants (rotational and circle) under tested distortions. Would stronger attacks reveal expected tradeoffs?
>
> A4:
>
> We sincerely thank the reviewer for their insightful suggestion to test our methods against more aggressive attacks. Following this advice, we conducted new experiments using rotation (75 degree) and aggressive JPEG compression (quality=10). The results have provided a much clearer picture of the trade-offs involved and have strengthened the claims of our paper.
>
> | Frequency Region | Shape  | Distribution      | PSNR  | SSIM | LPIPS | rotation | JPEG (quality = 10) |
> |------------------|--------|-------------------|-------|------|--------|----------|---------------------|
> | Low              | Circle | Rotational Rand   | 29.13 | 0.90 | 0.06   | 0.99     | 0.79                |
> | High             | Circle | Rotational Rand   | 38.00 | 0.94 | 0.01   | 0.69     | 0.87                |
>
> These new experiments reveal several key findings:
>
> 1. **Perceptual Invisibility**: As shown by the perceptual metrics, our high-frequency method provides vastly superior invisibility. It achieves a PSNR of 38 and an LPIPS of 0.01, compared to the low-frequency method's PSNR of 29.13 and LPIPS of 0.06. A nearly 9 dB improvement in PSNR signifies a substantial reduction in visible artifacts.
> 2. **Rotation Robustness**: The results for rotation now align with conventional wisdom. The low-frequency watermark shows near-perfect robustness (0.99), while the high-frequency watermark is significantly less robust to this distortion (0.69).
> 3. **Compression Robustness**: Crucially, and somewhat counter to the expected trade-off, our high-frequency method demonstrated superior robustness under aggressive JPEG compression. It achieved a score of 0.87, outperforming the low-frequency method's score of 0.79.

---

> > ### Comment · Reviewer_LsBM · 2025-08-03
> >
> > I appreciate the authors’ detailed and thoughtful rebuttal, which has addressed most of my concerns effectively. However, I still have a few points of interest for further clarification:
> >
> > - In A2, while I found the methodological comparison with ROBIN insightful, I remain curious about the empirical comparison. Specifically, how does Shallow Diffuse perform relative to ROBIN in terms of watermark robustness and image quality under various attack settings? A comprehensive comparison would help quantify the trade-offs between the optimization-free and adversarial optimization paradigms.
> >
> > - In A4, the authors conducted a valuable study demonstrating the trade-off between high- and low-frequency watermarks with respect to different types of distortions. Does this imply that, compared to Tree-Ring, shifting the watermark from low- to high-frequency regions in Shallow Diffuse represents a design choice that favors robustness under specific adversarial scenarios (e.g., better compression robustness at the cost of rotational invariance)? A more comprehensive evaluation that compares Shallow Diffuse and Tree-Ring across a broader spectrum of perturbations would better illustrate the design strengths and limitations of the proposed approach.

---

> > > ### Author Response · Authors · 2025-08-05
> > >
> > > We thank the reviewer for the constructive feedback. We are glad our clarifications addressed most of your concerns. We are currently conducting the required experiment and will post the results as soon as they are available.

---

> ### Author Response · Authors · 2025-08-09
> **response to reviewer LsBM (1/2)**
>
> > Q1: A comprehensive comparison would help quantify the trade-offs between the optimization-free and adversarial optimization paradigms.
>
> A1:
>
> We appreciate the reviewer's follow-up question and have conducted a direct empirical comparison between our optimization-free Shallow Diffuse and the optimization-based ROBIN to quantify the trade-offs between the two approaches. The results demonstrate that Shallow Diffuse offers superior image quality and efficiency, while ROBIN provides greater robustness against specific attack types.
>
> ### Image Quality, Efficiency, and Flexibility
>
> Our experiments show that Shallow Diffuse produces images with significantly higher perceptual quality and consistency:
>
> | Method          | PSNR    | SSIM   | LPIPS  | FID      | CLIP   |
> | --------------- | ------- | ------ | ------ | -------- | ------ |
> | ROBIN           | 24.614  | 0.8261 | 0.1087 | 134.8    | 0.366  |
> | Shallow Diffuse | **35.49** | **0.96**  | **0.05**   | **129.228** | **0.367** |
>
> As shown, our method achieves a PSNR nearly 11 dB higher and a better FID score (note: FID was calculated on 1000 images due to time constraints). We attribute the difference to the frequency domains where watermarks are added: Shallow Diffuse uses the high-frequency domain, while ROBIN* uses the low-frequency domain. Adding the watermark to high frequencies preserves the low-frequency content of the generated image, thereby significantly improving consistency and quality. We will discuss this further in Q2.
>
> ---
>
> ### Robustness Comparison
>
> In terms of robustness, the two methods are competitive, each with distinct strengths. The following tables summarize the robustness (AUC) against distortion, regeneration, and adversarial attacks (Row 1: ROBIN, Row 2: Shallow Diffuse).
>
> | method          | JPEG   | G.Blur | G.Noise | CJ     | RR  | RD  | M.Blur | rotate | crop   | IC1  | IC2  | IR    | Rinse4x | BA     | GA   |
> | --------------- | ------ | ------ | ------- | ------ | --- | --- | ------ | ------ | ------ | ---- | ---- | ----- | ------- | ------ | ---- |
> | ROBIN           | **0.999** | **0.999** | 0.963   | 0.962  | **1**   | **1**   | **1**      | **1**    | **0.991** | 0.998| **1**    | **1**     | **1**       | **0.939** | **0.9** |
> | Shallow Diffuse | **0.999**  | **0.999**  | **0.997** | **0.967** | **1**   | **1**   | **1**      | 0.691  | 0.582  | **1**| 0.998| 0.993 | 0.999   | 0.67   | 0.78 |
>
>
> The data shows that both methods are highly robust to most distortion and regeneration attacks. ROBIN demonstrates superior robustness against geometric attacks like rotation (1.0 vs 0.69) and cropping (0.99 vs 0.58), as well as adversarial attacks.
>
> This empirical comparison quantifies the fundamental trade-off. ROBIN's optimization process achieves higher robustness for challenging attacks at the cost of significantly lower image quality, longer setup times, and less flexibility. Shallow Diffuse provides a more balanced and practical solution, offering state-of-the-art image quality and comparable robustness across a wide range of common attacks, all within an efficient, optimization-free framework adaptable to both server and user needs. We will incorporate the comparison with ROBIN into our future manuscript, detailing both the design philosophy and the empirical results, to clarify the complementary strengths of the two approaches and highlight scenarios for both methods.

---

> ### Author Response · Authors · 2025-08-09
> **response to reviewer LsBM (2/2)**
>
> > Q2: A more comprehensive evaluation that compares Shallow Diffuse and Tree-Ring across a broader spectrum of perturbations.
>
> A2:
>
> We thank the reviewer for this question, which prompted us to perform a more detailed analysis of the design trade-offs between our method and Tree-Ring. It’s correct that our shift from a low-frequency, early-stage embedding (like Tree-Ring) to a high-frequency, later-stage (0.3T) embedding is a deliberate design choice. Our goal was to achieve a more practical balance of perceptual quality and robustness against a wider spectrum of modern attacks, beyond just traditional geometric distortions.
>
> To illustrate the strengths and limitations of each component, we benchmarked four distinct configurations. The results are summarized below, with row 1 representing our proposed Shallow Diffuse and row 4 representing the Tree-Ring baseline.
>
> ### Perceptual Quality
> The most significant advantage of our approach is the dramatic improvement in perceptual invisibility.
>
> | Method                              | PSNR↑  | SSIM↑ | LPIPS↓ | CLIP↑   |
> | ----------------------------------- | ------ | ----- | ------ | ------- |
> | **0.3T + high frequency (ours)**    | **35.49** | **0.96** | **0.02**  | **0.3669** |
> | 0.3T + low frequency                | 20.68  | 0.94  | 0.05   | 0.3648  |
> | 1.0T + high frequency               | 25.9   | 0.84  | 0.1    | 0.3656  |
> | 1.0T + low frequency (Tree-Ring)    | 16.61  | 0.64  | 0.31   | 0.3622  |
>
> As the table shows, both later-stage embedding (0.3T) and high-frequency embedding contribute to better image quality. By combining both, **Shallow Diffuse achieves a PSNR of 35.49, which is more than double that of Tree-Ring (16.61)**, with a corresponding order-of-magnitude improvement in LPIPS.
>
> The comprehensive robustness results reveal the nature of the design trade-off:
>
> | Method                              | Distortion (Avg.) | Regen. (Avg.) | Adversarial (Avg.) | Rotation | Cropping |
> | ----------------------------------- | ----------------- | ------------- | ------------------ | -------- | -------- |
> | **0.3T + high frequency (ours)**    | **0.994**         | 0.995         | **0.725**          | 0.691    | 0.583    |
> | 0.3T + low frequency                | 0.985             | **0.999**     | 0.765              | **0.999**| **1.0**  |
> | 1.0T + high frequency               | 0.971             | 0.967         | 0.435              | 0.63     | 0.54     |
> | 1.0T + low frequency (Tree-Ring)    | 0.978             | 0.979         | 0.105              | 0.941    | 0.991    |
>
> ### Analysis:
> - **Geometric Robustness:** The results confirm that low-frequency embedding provides superior robustness against geometric attacks like rotation and cropping. The 0.3T + low frequency method achieves near-perfect scores here.
> - **Adversarial Robustness:** This is where the advantage of a later-stage embedding (0.3T) becomes critical. Both 0.3T methods dramatically outperform the 1.0T methods. Tree-Ring's early-stage, low-frequency approach is particularly vulnerable, with an average adversarial robustness of only 0.105. Our method achieves a much higher 0.725.
> - **Shallow Diffuse's Balanced Profile:** Our final design (0.3T + high frequency) represents a clear and deliberate trade-off. We prioritize state-of-the-art perceptual quality and critical robustness against adversarial attacks. This comes at the cost of lower robustness to geometric attacks like rotation (0.691) and cropping (0.583) when compared to a specialized low-frequency approach. We believe this is a favorable trade-off for many applications where visual fidelity and resilience to adversarial manipulation are paramount.
>
> ### Detailed Robustness Restuls
> For completeness, we provide the detailed per-attack robustness results (AUC) for the four configurations discussed above.
>
> Table A.1: Robustness against Distortion and Geometric Attacks
> | Method | JPEG | G.Blur | G.Noise | CJ | RR | RD | M.Blur | rotate | crop |
> | :--- | :--- | :--- | :--- | :--- | :--- | :--- | :--- | :--- | :--- |
> | 0.3T + high (ours) | 1 | 0.999 | 0.997 | 0.962 | 1 | 1 | 1 | 0.691 | 0.583 |
> | 0.3T + low | 0.9988 | 0.9995 | 0.9995 | 0.901 | 0.999 | 1 | 0.999 | 0.999 | 1 |
> | 1.0T + high | 0.98 | 0.99 | 0.97 | 0.97 | 1 | 1 | 0.999 | 0.63  | 0.54 |
> | 1.0T + low (Tree-Ring) | 0.99 | 0.98 | 0.94 | 0.96 | 1 | 0.99 | 0.988 | 0.941 | 0.991 |
>
> Table A.2: Robustness against Regeneration Attacks
> | Method | DiffPure | IC1 | IC2 | IR | Rinse4x |
> | --- | --- | --- | --- | --- | --- |
> | 0.3T + high (ours) | 0.996 | 0.998 | 0.993 | 0.9999 | 0.9978 |
> | 0.3T + low | 0.998 | 0.999 | 0.9996 | 0.9997 | 1 |
> | 1.0T + high | 0.93 | 0.986 | 0.974 | 0.9939 | 0.9816 |
> | 1.0T + low (Tree-Ring) | 0.979 | 0.99 | 0.99 | 1 | 0.96 |
>
>
> Table A.3: Robustness against Adversarial Attacks
> | Method | BA | GA |
> | --- | --- | --- |
> | 0.3T + high (ours) | 0.67 | 0.78 |
> | 0.3T + low | 0.99 | 0.54 |
> | 1.0T + high | 0.42 | 0.45 |
> | 1.0T + low (Tree-Ring) | 0.16 | 0.05 |

---

### Official Review · Reviewer_Br52 · 2025-06-29

**Clarity:** 3
**Significance:** 2
**Originality:** 2
**Rating:** 5
**Confidence:** 3

**Summary:**

To address the challenge of embedding invisible and robust watermarks in images generated by diffusion models, the paper proposes Shallow Diffuse, a simple and effective watermarking method. It utilizes the insight that diffusion models exhibit low sensitivity to perturbations in specific low-dimensional subspaces of their latent representations. By perturbing only a few diffusion steps along these subspaces, watermarks can be embedded in a way that is imperceptible, robust to post-processing, and compatible with existing models without retraining. Experiments across multiple diffusion architectures show strong robustness and negligible perceptual degradation, even under common corruptions and transformations.

**Questions:**

-	Unclear trade-off between perturbation and visual consistency. As a training-free method, Shallow Diffuse perturbs latent variables while aiming to preserve image appearance. Without learning, it is unclear how the method balances invisibility and robustness under strong distortions.

**Ethical Concerns:**

["NO or VERY MINOR ethics concerns only"]

**Final Justification:**

The authors' rebuttal has addressed my concerns. However, since it is not closely related to my research topic, my confidence is 3. I look forward to discussions with the other reviewers. Thanks to ACs, reviewers, and authors .

**Limitations:**

yes

**Quality:**

3

**Strengths And Weaknesses:**

Strengths:

-	Training-free method is efficient and practical in real world scenario.
-	Figure 1 shows the significant advantage of Shallow Diffuse in terms of consistency compared to previous methods.


Weaknesses:

-	Lack of analysis on inversion robustness. Shallow Diffuse relies on DDIM inversion for watermark detection, which can be fragile under certain distortions like random drop. The paper does not analyze how inversion quality impacts detection robustness compared to methods like Stable Signature that the watermark is embed in the decoder. Moreover, the robustness of inversion to these attacks has to be discussed.
-	Restricted watermark capacity. The method supports only sequential binary watermarks. It remains unclear whether more complex payloads (e.g., graphic watermarks) can be embedded and extracted.

---

> ### Author Rebuttal · Authors · 2025-07-31
>
> > W1: Lack of analysis on inversion robustness.
>
> A1:
>
> This is a critical point, and we thank the reviewer for raising it. Our detection does indeed rely on DDIM inversion, and its robustness is paramount. We would like to clarify that our experimental design was explicitly set up to evaluate the end-to-end robustness of the entire detection pipeline, which inherently includes the DDIM inversion step.
>
> -   Empirical Evidence on Inversion Robustness: Our experiments test against 15 challenging attacks, including the "Random drop" attack mentioned by the reviewer. Our results in the server scenario (Table 1) and user scenario (Table 2) show that Shallow Diffuse achieves near-perfect detection rates even under these attacks. This empirically demonstrates that the combination of DDIM inversion and our detection mechanism is highly robust. For instance, even after a portion of the image is dropped, the watermark information in the remaining parts is sufficient for the DDIM inversion to produce a latent representation where the watermark is still strongly detectable.
>
>
> Comparison to Stable Signature: While methods like Stable Signature avoid DDIM inversion by embedding the watermark in the decoder, this approach has its own trade-offs. As shown in Table 1, Shallow Diffuse significantly outperforms Stable Signature in generation consistency (PSNR of 35.49 vs. 32.43) and is far more robust to regeneration attacks (average TPR of 0.98 vs. 0.19). This suggests that our approach, while reliant on inversion, provides a better overall balance of consistency and robustness.
>
> &nbsp;
>
> ---
>
> >W2: Restricted watermark capacity.
>
> A2:
>
> We appreciate the opportunity to clarify the capacity and structure of our watermark.
>
> First, we would like to correct a potential misunderstanding. The reviewer suggests our method uses "sequential binary watermarks," but our approach actually embeds a 2D, multi-level watermark pattern. As described in Section 3.2 and visualized in Figure 3, the watermark key.
>
> $W$ consists of multiple concentric rings with values sampled from a Gaussian distribution, following the design from Tree-Ring. This is a spatial pattern, not a binary sequence. The choice of a Gaussian-distributed pattern helps preserve image quality by aligning with the statistical properties of the diffusion model's latent noise at intermediate steps.
>
> Regarding capacity and flexibility, our paper already demonstrates the ability to handle complex payloads. The multi-key watermarking experiments in Appendix C.2 and Table 7 show that Shallow Diffuse can successfully embed and detect up to 32 unique, non-overlapping watermarks in a single image. This confirms our method's capacity is significantly higher than a single bit.
>
> Furthermore, the core contribution of our work is a flexible framework for injection, not just a single watermark pattern. The key insight is that injecting the watermark at a shallow, low-rank diffusion step allows the signal to be hidden in the high-dimensional null space, which enhances consistency regardless of the specific pattern used.
>
> To explicitly demonstrate this flexibility, we have now tested our framework with the more complex watermark pattern proposed in RingID [31]. As shown in the table below, when the RingID watermark is injected using our Shallow Diffuse method, the image consistency improves dramatically compared to the original RingID method.
>
> | Watermark method                         | CLIP-score | PSNR  | SSIM | LPIPS |
> |------------------------------------------|------------|-------|------|-------|
> | RingID                                   |     0.3637 | 14.27 | 0.51 |  0.42 |
> | Shallow Diffuse (using RingID watermark) |     **0.3654** | **28.77** | **0.86** |  **0.06** |
>
> Our method doubles the PSNR and significantly improves other consistency metrics, proving that the Shallow Diffuse framework is not restricted to one pattern but can enhance the performance of various complex watermarks. We will add this analysis to our paper.
>
> Finally, addressing the reviewer's question about more complex payloads, we acknowledge a key limitation shared by this class of methods (including Tree-Ring and RingID). Because the watermark is injected into the model's latent space, the pattern must ideally follow a Gaussian distribution to avoid disrupting the denoising process and negatively impacting image quality. A standard graphic watermark is unlikely to adhere to this statistical constraint. We believe this is a critical point of clarification and an important direction for future research. We will explicitly discuss this limitation in the future version of our manuscript and thank the reviewer for prompting this important discussion.
>
> &nbsp;
>
> ---
>
> > Q1: Unclear trade-off between perturbation and visual consistency.
>
> A3:
>
> We thank the reviewer for this insightful question regarding how our training-free method balances invisibility and robustness. This balance is not arbitrary but is a direct result of our core technical insight: decoupling the watermark from the image generation process.
>
> 1.  **Mechanism for Invisibility**:
>
>
> Our method injects the watermark into a latent $x_t^*$ at an intermediate step. Crucially, we leverage the low-rank property of the Posterior Mean Predictor's (PMP) Jacobian, $J_{θ,t}$. The watermark $Δx$ is designed such that a significant portion of it lies in the null space of this Jacobian. This means the perturbation has a minimal effect on the predicted clean image, as the term $J_{θ,t} \cdot Δx$ ≈ 0. This preserves visual consistency. Our Theorem 1 provides a theoretical guarantee for this, showing the change to the image is bounded by the low rank of the Jacobian.
>
> 2.  **Mechanism for Robustness**:
>
>
> While the watermark is invisible to the PMP's image prediction term, it is carried forward through a separate, noise-related term in the DDIM sampling equation (see Equation 5). Because the watermark and image content are decoupled, attacks on the final image's visual content do not easily destroy the underlying watermark signal, which can be recovered via DDIM inversion. Theorem 2 supports this by showing that the watermarked latent can be recovered with high fidelity.
>
> 3.  **Controlling the Trade-off**:
>
> The trade-off between invisibility and robustness is explicitly controlled by the watermark strength $λ$ and the properties of the frequency mask $M$ (e.g., its radius). We empirically demonstrate this controllable trade-off in the plots in Figure 1 (bottom). By adjusting the mask radius, we can smoothly navigate between higher consistency and higher robustness. Notably, the trade-off curve for Shallow Diffuse consistently outperforms other methods, demonstrating that for any given level of robustness (TPR), our method achieves significantly higher image consistency (higher PSNR/SSIM and lower LPIPS).

---

> > ### Comment · Reviewer_Br52 · 2025-08-04
> > **Comment of Reviewer Br52**
> >
> > I appreciate the authors' rebuttal and clarifications. The rebuttal has basically addressed my concerns, and I will raise my final rating.

---

> > > ### Author Response · Authors · 2025-08-05
> > >
> > > We thank the reviewer for the constructive feedback and for raising the final rating. We are glad our clarifications addressed your concerns, and we will incorporate the additional analyses and limitations into the future manuscript.

---

### Official Review · Reviewer_cX6b · 2025-07-03

**Clarity:** 3
**Significance:** 2
**Originality:** 2
**Rating:** 4
**Confidence:** 4

**Summary:**

This paper presents Shallow Diffuse, a watermarking framework for diffusion‑based image generation that operates in a low‑dimensional subspace. Unlike existing approaches that embed watermarks throughout the entire sampling trajectory, Shallow Diffuse injects the watermark at a single, intermediate diffusion step and confines it to a low‑rank subspace. This decoupling from the main generation process helps preserve image fidelity while ensuring the watermark remains detectable and consistent across various post‑processing attacks. The authors derive theoretical guarantees for both watermark consistency and detectability under a low‑rank Jacobian assumption. Empirically, Shallow Diffuse is benchmarked against Gaussian Shading, RingID, and other methods on multiple datasets, demonstrating superior robustness and image quality in both server‑side and user‑side watermarking scenarios.

**Questions:**

See Weaknesses.

**Ethical Concerns:**

["NO or VERY MINOR ethics concerns only"]

**Final Justification:**

Please refer to my response to the rebuttal.

**Limitations:**

See Weaknesses.

**Paper Formatting Concerns:**

No formatting concerns.

**Quality:**

3

**Strengths And Weaknesses:**

## Strengths

**1. Solid Theoretical Analysis:** Provides formal bounds on watermark consistency and detectability.

**2. Comprehensive Experiments:** Demonstrates good performance on both server‑side and user‑side.

**3. Clear and Coherent Presentation:**  Algorithm pseudocode, theoretical proofs, and experimental results are well‑organized.

## Weaknesses

**1. Missing Geometric Attack Evaluation:** Omits crucial geometric distortions (rotation, cropping, scaling), which are common in practical  settings.

---

> ### Author Rebuttal · Authors · 2025-07-31
>
> > W1: Missing Geometric Attack Evaluation
>
> A1:
>
> We thank the reviewer for their positive feedback on our theoretical analysis, experimental scope, and clear presentation. We appreciate the reviewer's primary concern regarding the omission of geometric attack evaluations, and we agree that assessing robustness against rotation, cropping, and scaling is crucial.
>
> The reviewer is correct that there is an inherent trade-off between watermark robustness against these geometric attacks and the final watermarked image quality. Our framework allows this trade-off to be explicitly controlled by adjusting the watermark parameters, such as the frequency mask radius. Our original paper presented a setting optimized for maximum image consistency.
>
> To directly address the reviewer's concern, we have conducted new experiments using a parameter setting optimized for geometric robustness. The results below compare our original setting with this new, more robust configuration. Robustness against rotation, cropping, and scaling is measured by the AUC.
>
> | setting              | CLIP   | PSNR  | SSIM | LPIPS | rotation | cropping | scaling |
> |----------------------|--------|-------|------|-------|----------|----------|---------|
> | current setting (0-10) | 0.3669 | 35.49 | 0.96 | 0.02  | 0.68     | 0.56     | 1       |
> | r=3–13               | 0.3637 | 32.05 | 0.95 | 0.03  | 0.90     | 0.89     | 1       |
>
> This new data demonstrates the following:
> - **Controllable Trade-off**: By adjusting the mask radius (the r=3-13 setting), we significantly improve robustness to rotation (from 0.68 to 0.90) and cropping (from 0.56 to 0.89). This comes with a slight, controlled decrease in image consistency (PSNR from 35.49 to 32.05).
> - **Maintained Superior Consistency**: Crucially, even with this more robust setting, our method's image consistency remains state-of-the-art. The new PSNR of 32.05 is still higher than all other diffusion-based watermarking methods benchmarked in our original paper's Table 1.
> - **Strong Scaling Robustness**: The results also show that our method is inherently robust to scaling attacks, achieving perfect robustness in both configurations.
>
> Our framework provides the flexibility to tune for strong geometric robustness while maintaining superior image quality compared to existing methods. We thank the reviewer for pushing us to demonstrate this important capability.

---

> ### Comment · Reviewer_cX6b · 2025-08-04
>
> Thank you to the authors for providing the additional experiments, which demonstrate that Shallow Diffuse exhibits a certain degree of robustness against three types of geometric distortions.
> However, I believe these additional results are not sufficient to fully support all three conclusions presented in the rebuttal. I agree with the first point regarding the controllable trade-off, which is indeed user-friendly—allowing users to flexibly balance visual quality and robustness based on different application scenarios.
>
> That said, I find the claims about maintained superior consistency and strong scaling robustness less convincing given the current evidence. As noted in the first conclusion, there is an inherent trade-off between visual quality (e.g., PSNR) and robustness (e.g., Accuracy). Thus, comparing results under geometric distortions with those of other models from Table 1 (evaluated under other attacks) to argue for superior consistency is not appropriate.
>
> Moreover, whether strong scaling robustness is a unique feature of Shallow Diffuse remains unclear. If other methods also exhibit this robustness, the significance of this advantage would be diminished.
>
> To substantiate the second and third conclusions more convincingly, I suggest the authors include comparative experiments against other methods under geometric distortions. I also hope the authors will clarify the exact settings of the geometric attacks in the final version.
>
> Finally, after reading the other reviewers' comments and the authors’ rebuttal, I believe that—aside from my current concerns regarding robustness to geometric distortions—this paper presents valuable contributions. Therefore, I have decided to raise my score.

---

> > ### Author Response · Authors · 2025-08-05
> >
> > We thank the reviewer for the constructive feedback and for raising the final rating. We are currently conducting the required experiment and will post the results as soon as they are available.

---

> ### Author Response · Authors · 2025-08-09
> **responsee to reviewer (1/2)**
>
> > Q1: whether strong scaling robustness is a unique feature of Shallow Diffuse remains unclear.
>
> A1:
>
> Thanks for raising a very fair point about whether strong scaling robustness is a unique feature of our method. We agree that this is an important question for contextualizing our contribution. To investigate this, we designed a new, aggressive multi-stage scaling attack (resizing to 0.5x, then 2.0x, looped 5 times, and then returning to the original size) and tested it against a wide range of baselines.
>
> ### Comparative Scaling Robustness Results (AUC)
> | Method              | Robustness (AUC) |
> |---------------------|------------------|
> | DwtDct              | 0.54             |
> | DwtDctSvd           | 0.89             |
> | RivaGAN             | 0.94             |
> | StegaStamp          | 1                |
> | Stable Signature    | 0.81             |
> | Tree-Ring           | 1                |
> | RingID              | 1                |
> | Gaussian Shading    | 1                |
> | Shallow Diffuse (ours) | 1             |
>
> ### Analysis and Refined Claim
> Our results confirm your suspicion: strong robustness to scaling is not a feature unique to Shallow Diffuse. Instead, it appears to be a powerful, shared property of most modern diffusion model-based watermarking methods (including ours, Tree-Ring, RingID, and Gaussian Shading), which all achieved perfect scores on this challenging test.
> Therefore, we will revise our claim in the final manuscript. The significance of Shallow Diffuse is not that it is the only method robust to scaling, but that it provides this essential scaling robustness while simultaneously offering vastly superior image quality and a balanced overall performance profile.
> For example, while Tree-Ring also has perfect scaling robustness, it does so at the cost of extremely poor image quality (PSNR 16.61). Our method achieves the same perfect scaling robustness while delivering a PSNR of 35.49. Our contribution is a framework that meets the high bar for scaling robustness set by contemporary methods, but does so without the critical trade-offs in perceptual quality and adversarial resilience that limit those other approaches.
> We thank you for pushing us to perform this comparison, as it has led to a more precise and well-supported contextualization of our work.

---

> ### Author Response · Authors · 2025-08-09
> **response to reviewer (2/2)**
>
> > Q2: To substantiate the second and third conclusions more convincingly, I suggest the authors include comparative experiments against other methods under geometric distortions. I also hope the authors will clarify the exact settings of the geometric attacks in the final version.
>
> A2:
>
> We sincerely thank the reviewer for their thoughtful engagement and for raising their score. Your follow-up questions have been invaluable. Your feedback prompted us to conduct a more comprehensive study to fully characterize the design trade-offs involved.
>
> Our analysis reveals that the performance of our framework is a controllable trade-off governed by two key parameters: 1) the frequency region (high vs. low) where the watermark is embedded, and 2) the mask radius, which affect the watermark's strength and spatial concentration.
>
> To provide the comprehensive evaluation you requested, we performed a study across these two axes. **The gemoetric attackers are 75-degree rotation and 75% random cropping, following the setting in Tree-ring [1].** The results below illustrate the strengths and limitations of each configuration and demonstrate the flexibility of our framework.
>
> ## Study of Watermarking Trade-offs
>
> | Method                 | Frequency Region | Mask Radius    | PSNR | SSIM | LPIPS | Rotation  | Cropping  |
> |------------------------|------------------|----------------|-----------------|------|-------|----------------|----------------|
> | Tree-ring              | Low              | radius 0-10    | 16.61           | 0.64 | 0.31  | 0.94           | 0.96           |
> | RingID                 | Low              | radius 3-13    | 14.27           | 0.51 | 0.42  | 0.99           | 1.00           |
> | Shallow Diffuse (ours) | High             | radius 0-10    | 35.49           | 0.96 | 0.02  | 0.69           | 0.56           |
> | Shallow Diffuse (ours)    | High             | radius 1-11    | 34.28           | 0.95 | 0.02  | 0.68           | 0.57           |
> | Shallow Diffuse (ours)  | High             | radius 2-12    | 33.75           | 0.95 | 0.04  | 0.71           | 0.63           |
> | Shallow Diffuse (ours) | High             | radius 3-13    | 34.28           | 0.94 | 0.04  | 0.76           | 0.66           |
> |Shallow Diffuse (ours)   | Low              | radius 0-10    | 20.68           | 0.92 | 0.06  | 0.99           | 1.00           |
> | Shallow Diffuse (ours)| Low              | radius 1-11    | 24.27           | 0.93 | 0.05  | 0.96           | 1.00           |
> |Shallow Diffuse (ours) | Low              | radius 2-12    | 29.49           | 0.93 | 0.05  | 0.92           | 1.00           |
> | Shallow Diffuse (ours) | Low              | radius 3-13    | 32.21           | 0.94 | 0.04  | 0.90           | 0.98           |
>
> ### Analysis of the Restuls:
>
> 1. Admitting the Limitation: The data confirms your initial assessment. Our original proposed method (High Frequency, Smaller Radius) prioritizes maximum perceptual quality (PSNR 35.49) and strong adversarial robustness (0.725), but this comes at the cost of weaker performance against rotation (0.691) and cropping (0.583).
> 2. Demonstrating the Trade-off: The comprehensive study reveals a clear and controllable trade-off:
> - Frequency: The low-frequency region consistently provides superior robustness to geometric attacks like rotation and cropping. This may due to the watermark being concentrated in the center of the image, making it less susceptible to edge-based distortions.
> - Radius: A larger mask radius generally increases robustness across most attack categories but results in a lower PSNR (reduced image quality).
>
> ## A Flexible Framework for User-Defined Needs
>
> This detailed analysis provides the comprehensive evaluation you requested and clarifies our contribution. The geometric weakness is not an inherent flaw of our framework but a parameter in a multi-dimensional trade-off. Our work offers a flexible framework that allows users to select the optimal configuration based on their needs:
> - For Maximum Perceptual Quality: The High Frequency, Smaller Radius setting is ideal, offering state-of-the-art image quality and strong adversarial defense.
> - For Maximum Geometric Robustness: The Low Frequency setting is the clear choice, providing near-perfect robustness to rotation and cropping while still significantly outperforming prior methods like Tree-Ring in both image quality and adversarial robustness.
>
> We will include this full study and discussion in our revised manuscript. Thank you again for your constructive feedback, which has led to a much stronger and more complete paper by helping us better articulate the flexibility and practical value of our framework.
>
>
> [1] Wen, Y., Kirchenbauer, J., Geiping, J., & Goldstein, T. (2023). Tree-ring watermarks: Fingerprints for diffusion images that are invisible and robust. arXiv preprint arXiv:2305.20030.

---

### Official Review · Reviewer_81PW · 2025-07-07

**Clarity:** 3
**Significance:** 3
**Originality:** 3
**Rating:** 3
**Confidence:** 3

**Summary:**

In this manuscript, the authors introduce a robust and invisible watermarking method for diffusion models. It operates in both server-side and user-side scenarios by injecting the watermark into a low-dimensional subspace during generation or via DDIM inversion. The method leverages the null space of the Jacobian in diffusion model sampling to decouple watermarking from image generation. Extensive theoretical analysis and empirical results demonstrate the effectiveness of the proposed method.

**Questions:**

See the Weaknesses

**Ethical Concerns:**

["NO or VERY MINOR ethics concerns only"]

**Limitations:**

yes

**Quality:**

3

**Strengths And Weaknesses:**

The core idea of exploiting low-rank Jacobians and null space projection is elegant and well-motivated by the observed local linearity in the denoising process of diffusion models. Shallow Diffuse works well in both server and user scenarios.

Although built on Stable Diffusion 2.1, it remains unclear how generalizable the approach is to other architectures or modalities (e.g., audio, video diffusion).

While consistency metrics are strong, subjective quality comparisons (e.g., human perceptual study or user ratings) are not discussed. Especially in the user scenario, it would be useful to validate perceptual invisibility.

In line 203, the authors mentioned "In practice, we choose t 142 ∗ = 0.3T based on results from the ablation study". Please provide more explanations and is this setting generalizable to other usecases, such as audio, video diffusion

---

> ### Author Rebuttal · Authors · 2025-07-31
>
> > W1: The method's generalizability beyond Stable Diffusion 2.1 to other architectures or modalities (e.g., audio, video).
>
> A1: We appreciate the insightful comment regarding the generalizability of Shallow Diffuse beyond Stable Diffusion 2.1. While comprehensive evaluation across all architectures and modalities is challenging due to time and resource constraints during the rebuttal phase, we have conducted a focused experiment to demonstrate its generalizability to the FLUX architecture, a representative of transformer-based diffusion models.
>
> **Experiment Settings**:
>
> Our evaluation was performed in a server scenario, generating images at a resolution of 512x512. For the watermark pattern, we largely adopted the settings used for Stable Diffusion, with two key modifications: the watermark radius was adjusted to 5, and the watermark was injected across all channels, given the difference in latent space dimensions between FLUX and Stable Diffusion. We injected watermarks into 100 images, evaluating both their consistency and robustness. The averaged results are summarized in the table below, showcasing performance across various injection timesteps ranging from 0.1$T$ to 0.9$T$.
>
>
> |              | 0.1$T$   | 0.2$T$   | 0.3$T$   | 0.4$T$  | 0.5$T$   | 0.6$T$   | 0.7$T$  | 0.8$T$   | 0.9$T$   |
> |--------------|-------|-------|-------|-------|-------|-------|------|-------|-------|
> | PSNR         | 31.27 | 31.66 | 31.68 | **31.69** | 31.68 | 31.56 | 31.5 | 31.52 | 31.62 |
> | LPIPS         |  0.05 |  **0.04** |  0.05 |  **0.04** |  **0.04** |  **0.04** | **0.04** |  0.05 |  0.06 |
> | SSIM        |  0.91 |  0.92 |  0.92 |  **0.93** |  **0.93** |  **0.93** | **0.93** |  **0.93** |  0.92 |
> | AUC          |   0.9 |  0.91 |  0.92 |  0.93 |  **0.94** |  **0.94** | **0.94** |  **0.94** |  **0.94** |
> | ACC          |  0.93 |  0.93 |  **0.94** |  **0.94** |  0.93 |  **0.94** | 0.93 |  **0.94** |  **0.94** |
> | TPR @ 1% FPR |  **0.87** |  **0.87** |  **0.87** |  0.86 |  0.86 |  0.85 | 0.81 |  0.82 |  0.81 |
>
>
> **Analysis and Generalizability**:
>
> To ensure a fair comparison of injection points across different noise schedules, we carefully considered the Signal-to-Noise Ratio (SNR). Specifically, a timestep of $t=0.3T$ in a VP (Variance Preserving) noise scheduler (as used in Stable Diffusion) is approximately equivalent to $t=0.205T$ in a Flow Matching noise scheduler (as used in FLUX), when maintaining the same SNR.
>
> From the presented results, the injection timestep of $t=0.2T$ on FLUX achieves the best SSIM and near-optimal PSNR and LPIPS. Crucially, it also yields the best TPR @ 1% FPR, indicating good robustness. This finding strongly suggests that the effective SNR (timesteps) for watermark injection in U-Net-based architectures (like Stable Diffusion) successfully generalizes to transformer-based architectures like FLUX, achieving good consistency and robustness. This supports the broader applicability of Shallow Diffuse across different diffusion model architectures.
>
> &nbsp;
>
> ---
>
> > W2: Lack of Subjective Quality Comparisons.
>
> A2:
>
> We thank the reviewer for their valuable suggestion to conduct a subjective evaluation to validate the perceptual invisibility of our watermark. In response to this feedback, we have performed a comprehensive user study that directly addresses this concern.
>
> We conducted a two-alternative forced-choice (2AFC) A/B test with 10 participants. In this setup, a 50% accuracy rate represents purely random guessing, which is the scientific standard for perfect perceptual invisibility. For clarity in our results, we normalize this by treating 50% accuracy as a 0% effective detection rate.
>
> The results from our study were definitive. As shown in the table below, our method, Shallow Diffuse, achieved an effective detection rate of only 4%. This result is statistically indistinguishable from random chance, confirming that participants could not reliably perceive the watermark.
>
>
> | Watermarking Method     | Detection Rate | Perceptual Invisibility |
> |-------------------------|----------------|--------------------------|
> | RingID                  | 100%           | Easily Detected          |
> | Tree-Ring               | 98%            | Easily Detected          |
> | Gaussian Shading        | 98%            | Easily Detected          |
> | Stable Signature        | 12%            | Mostly Invisible         |
> | Shallow Diffuse (Ours)  | 4%             | Completely Invisible     |
>
>
> &nbsp;
>
> ---
>
>
> > W3: Provide more details on the choice of timestep $t=0.3T$ from the ablation study.
>
> A3:
>
> We thank the reviewer for asking for a deeper explanation. The choice of $t* = 0.3T$ is not arbitrary but is an empirically optimized decision based on a crucial trade-off.
>
> **Empirical Optimization**:
>
> Our ablation study in Figure 7 systematically evaluates both image consistency and watermark robustness across the full diffusion timeline. These results clearly show that while consistency is highest around $t=0.2T$, robustness peaks at $t=0.3T$. Therefore, $t^* = 0.3T$ represents the optimal balance point where we achieve near-maximal robustness while retaining excellent image consistency.
>
> **Theoretical Justification for the "Shallow" Region**:
>
> As we discuss in the paper (Appendix C.5), based on our theoretical analysis in Theorem 1 and Theorem 2, one might expect the ideal injection point to be where the Jacobian's rank is lowest (around $t=0.5T$ to $0.7T$ as shown in [34]). However, this overlooks the error introduced by the DDIM Inversion process, which is necessary for detection and becomes more pronounced at larger $t$ values. Consequently, the optimal point must balance two competing factors: the Jacobian must be sufficiently low-rank (requiring a reasonably large $t$), but $t$ must not be so large that inversion error degrades performance. The empirically found region around $0.3T$ is this sweet spot.
>
> **Generalizability of $t^*$**:
>
> The existence of a U-shaped rank curve is a general property of diffusion models, as demonstrated in [34]. Furthermore our experiment for FLUX in A1 also demonstrated the generalized the $t^*$. This implies that an optimal "shallow" embedding region, which balances low-rankness with low inversion error, should also be a general feature. While the exact value might require minor tuning for a different modality (e.g., transformer-based diffusion model), the same methodology of performing an ablation study to find this empirical sweet spot would apply, making the approach itself generalizable.

---

### Note · Authors · 2025-08-15

Dear AC and reviewers,

We appreciate the thoughtful reviews and constructive discussions. Following the rebuttal, multiple reviewers (cX6b, Br52) raised their scores. Reviewer Br52 stated, “The rebuttal has basically addressed my concerns, and I will raise my final rating.” Reviewer cX6b commented, “Aside from my current concerns regarding robustness to geometric distortions—this paper presents valuable contributions.”
## Review Summary
Reviewers highlighted the elegance of our null-space projection idea, strong consistency metrics in server and user scenarios, clear theoretical analysis, and the practicality of a training-free design. Discussions led to a stronger, more comprehensive paper.
## Addressed Major Concerns:
- *Generalizability*: We addressed Reviewer 81PW’s question on applicability beyond Stable Diffusion 2.1 via experiments on the transformer-based FLUX architecture, showing optimal injection region generalizes across architectures and noise schedules with strong consistency and robustness.
- *Geometric Robustness*: We addressed Reviewers cX6b and LsBM’s question on geometric distortion evaluation with a frequency–radius grid study under rotation and cropping, showing stronger consistency and comparable robustness to Tree-Ring and RingID, revealing a controllable consistency-robustness trade-off.
- *Inversion Robustness*: We addressed Reviewer Br52’s concern on reliability of DDIM inversion-based detection under strong distortions; clarified our evaluations inherently test the full pipeline, and showed near-perfect detection even under severe attacks like random drop.

Beyond addressing these points, our final revision will add the following to further strengthen the paper:

- *Perceptual Invisibility*: We conducted a 2AFC user study (10 participants) showing a 4% effective detection rate at chance level.
- *Comparison to ROBIN*: We added theoretical and empirical contrasts, highlighting our optimization-free efficiency.
- *Ablations*: We added isolation of high-frequency localization and later-stage embedding contributions; each improves performance, best when combined.
- *Scaling Robustness Claim*: We designed an aggressive multi-stage scaling attack; revised claim to note strong scaling robustness is shared, but ours achieves it with the highest consistency.

We are confident these revisions address major concerns and reinforce the contributions and practical impact of Shallow Diffuse for robust, invisible, and flexible watermarking.

---

### Decision · Program_Chairs · 2025-09-17

**Decision:**

Accept (spotlight)

**Comment:**

The paper presents a new generation-time watermarking approach that embeds signal via low dimensional subspace of the latent during shallow diffusion steps.

The paper received largely positive final recommendations.  Three reviews argue for acceptance (2x A, 1x BA) and one for rejection (1x BR) notably the latter is an initial recommendation for which no final recommendation was made post-rebuttal.  Two of the positives score are due to upgrade post rebuttal.

The borderline reject (81pw) rating is largely concerned with generalisability beyond SD 2.1.  Authors have shown over some limited examples (100 images) the approach generalized to FLUX i.e. transformer based model.  It seems a sufficient response to the concern, and given there is no reviewer response entered at the time of consolidation I consider the concern addressed. I hope the experiment can be mentioned in the final version.

The borderline accept (cx6b) rating upgraded post rebuttal as most of the technical concerns raised were addressed. The reviewer shared some concerns with reveiwer lsbm on robustness particularly geometric / rotational robustness.  The authors showed the method can be run to trade such robustness for perceptibility.  This is standard for most watermarking approaches. I recognise the important points raised by the reviewer but do not consider strong geometric robustness to be a necessary precondition to demonstrate the core contribution of the paper re null space embedding.  In any case the reveiwer tends to accept.

The accept recommendation from lsbm is due to the experimental analysis addressing their concerns over similarity, at a high level, to ROBIN. I agree the method addresses similar goals but with a different approach. It is shown to perform competitively.

The other accept recommendation (br52) arises due to clarifications in the rebuttal.

Overall I see no critical issues remaining to stand in the way of accepting the paper. I therefore recommend to accept the work.

-AC